# Human CD26^{high} T cells elicit tumor immunity against multiple malignancies via enhanced migration and persistence

Stefanie R. Bailey[1,2,3], Michelle H. Nelson [1,2,3,7], Kinga Majchrzak[1,2,3,8], Jacob S. Bowers[1,2,3], Megan M. Wyatt [1,2,3], Aubrey S. Smith[1,2,3], Lillian R. Neal[1,2,3], Keisuke Shirai[4,9], Carmine Carpenito[5,10], Carl H. June [5], Michael J. Zilliox[6] & Chrystal M. Paulos[1,2,3]

CD8[+] T lymphocytes mediate potent immune responses against tumor, but the role of human CD4[+] T cell subsets in cancer immunotherapy remains ill-defined. Herein, we exhibit that CD26 identifies three T helper subsets with distinct immunological properties in both healthy individuals and cancer patients. Although CD26^{neg} T cells possess a regulatory phenotype, CD26^{int} T cells are mainly naive and CD26^{high} T cells appear terminally differentiated and exhausted. Paradoxically, CD26^{high} T cells persist in and regress multiple solid tumors following adoptive cell transfer. Further analysis revealed that CD26^{high} cells have a rich chemokine receptor profile (including CCR2 and CCR5), profound cytotoxicity (Granzyme B and CD107A), resistance to apoptosis (c-KIT and Bcl2), and enhanced stemness (β-catenin and Lef1). These properties license CD26^{high} T cells with a natural capacity to traffic to, regress and survive in solid tumors. Collectively, these findings identify CD4[+] T cell subsets with properties critical for improving cancer immunotherapy.

[1] Department of Microbiology and Immunology, Medical University of South Carolina, Charleston, SC 29425, USA. [2] Department of Surgery, Medical University of South Carolina, Charleston, SC 29425, USA. [3] Department of Dermatology and Dermatologic Surgery, Medical University of South Carolina, Charleston, SC 29425, USA. [4] Hematology/Oncology Division, Hollings Cancer Center, Medical University of South Carolina, Charleston, SC 29425, USA. [5] Department of Pathology and Laboratory Medicine, University of Pennsylvania Cancer Center, Philadelphia, PA 19104, USA. [6] Department of Public Health Sciences, Stritch School of Medicine, Loyola University Chicago, Maywood, IL 60153, USA. [7] Aptevo Therapeutics, Seattle, WA 98121, USA. [8] Present address: Department of Physiological Sciences, Faculty of Veterinary Medicine, Warsaw University of Life Sciences, Warsaw, 02-787, Poland. [9] Department of Medicine, Geisel School of Medicine, Dartmouth College, Hanover, NH 02714, USA. [10] Present address: Eli Lilly and Company, New York, NY 10016, USA. Correspondence and requests for materials should be addressed to S. R.B. (email: flemins@musc.edu) or to C. M.P. (email: paulos@musc.edu)

Cancer patients have been treated with various therapies and until recently, many with poor outcomes. The discovery of cell-intrinsic inhibitory pathways and cancer-specific antigens has allowed for the advancement of immune checkpoint blockades[1, 2] and a cellular therapy called adoptive cell transfer (ACT), respectively. ACT is an innovative therapy that entails the acquisition, expansion and infusion of autologous T cells back into the patient to eradicate tumors[3]. The ability to engineer T cells with T cell receptors (TCRs[4, 5]) or chimeric antigen receptors (CARs[6, 7]) has made this therapy available to more individuals. Despite the impressive results of CAR-T therapy in patients with blood-based malignancies, it has yielded poor results in patients with solid tumors thus far[8, 9].

Although tumor-infiltrating lymphocytes (TILs[10, 11]) or immune checkpoint modulators[12, 13] regress malignancies in some patients bearing immunogenic solid tumors, these approaches have been ineffective at treating poorly immunogenic tumors such as mesothelioma and pancreatic cancer[14, 15]. Though several factors could have a role in why these therapies fail, two possible characteristics crucial for effective tumor clearance include the ability of T cells to traffic to[16, 17] and persist in the tumor[18, 19]. Although CD8+ T cells have shown clinical promise[20] and the capacity to repopulate[21], human CD4+ T cell subsets that exhibit properties of stemness and natural migration to the tumor have yet to be identified.

Previous work on CD4+ T cells has shown that cells polarized to a type 17 phenotype—Th17 cells—exhibit stem cell-like qualities and yield greater tumor regression and persistence in vivo than other traditional T helper subsets[22, 23]. However, the

expansive culture conditions required to generate these cells in vitro has inhibited their transition to the clinic. Recently, Bengsch et al.[24] reported that human T cells with a high expression of CD26 on their cell surface—termed CD26high T cells—produce large amounts of the Th17 hallmark cytokine, IL-17. CD26 is an enzymatically active, multi-functional protein shown to have a role in T cell costimulation as well as the binding of extracellular matrix proteins/adenosine deaminase[25]. Despite being well studied in autoimmune diseases such as diabetes[26], the role of CD26 and its enzymatic activity in cancer has yet to be fully explored. Given the substantial IL-17 production from CD26high T cells, we postulated that CD26 expression on CD4+ T cells might correlate with a more stem cell-like lymphocyte with enhanced tumor regression.

Herein, we report that CD26 distinguishes three distinct human CD4+ subsets with varying responses to human tumors: one with regulatory characteristics (CD26neg), one with a naive phenotype (CD26int), and one with properties of durable memory and stemness (CD26high). CD26high T cells persist and regress/control tumors to a far greater extent than CD26neg T cells and surprisingly, slightly better than naive CD26int T cells. Our data reveal that CD26high T cells have enhanced multi-functionality (IL-17A, IFNγ, IL-2, TNFα, and IL-22), stemness properties (elevated β-catenin and Lef1), memory (long-term persistence and Bcl2 expression), and a rich profile of chemokine receptors (including CCR2 and CCR5), thereby enabling them to traffic to, regress mesothelioma and inhibit the growth of pancreatic tumors. Furthermore, better antitumor responses correlate with an increased presence of CD26+ T cells in

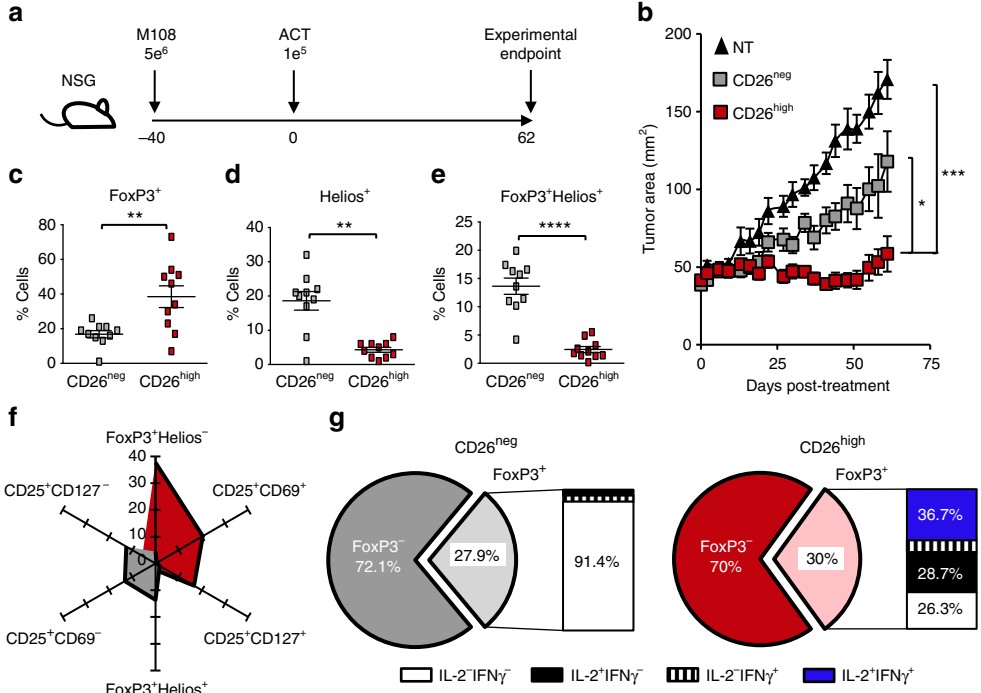

**Fig. 1** Human CD26high T cells are activated and display antitumor activity. CD4+ lymphocytes were isolated from healthy donor PBMCs, sorted by CD26 expression and stimulated with magnetic beads coated with CD3 and ICOS agonists (cultured at a ratio of 1 bead to every 5 T cells). T cells were transduced 36 h post-activation with a lentiviral vector encoding a first generation chimeric antigen receptor that recognizes mesothelin and stimulates the CD3ζ domain. These cells were expanded for 10 days with IL-2 (100 IU/ml). **a, b** NSG mice were subcutaneously injected with 5e6 M108 mesothelioma cells. Forty days post-M108 establishment, mice were intravenously infused with 1e5 human CD26neg or CD26high T cells redirected to express MesoCAR. Tumors were measured bi-weekly (N = 10 mice per group). P values for the tumor curve were calculated by one-way ANOVA with a Kruskal-Wallis comparison using final tumor measurements from day 62. **c–g** Graphical representations of transcription factors (**c–f** N = 10) and cytokine production (**g** N = 3) by sorted T cells isolated from multiple healthy individuals prior to bead stimulation. In **g**, the frequency of FoxP3+ cells from the enriched CD26neg or CD26high cultures secreting inflammatory cytokines were assayed by flow cytometry. P values for **c–e** were calculated using a Mann-Whitney U Test. Data with error bars represent mean ± SEM. *P < 0.05; **P < 0.01; ***P < 0.001; ****P < 0.0001

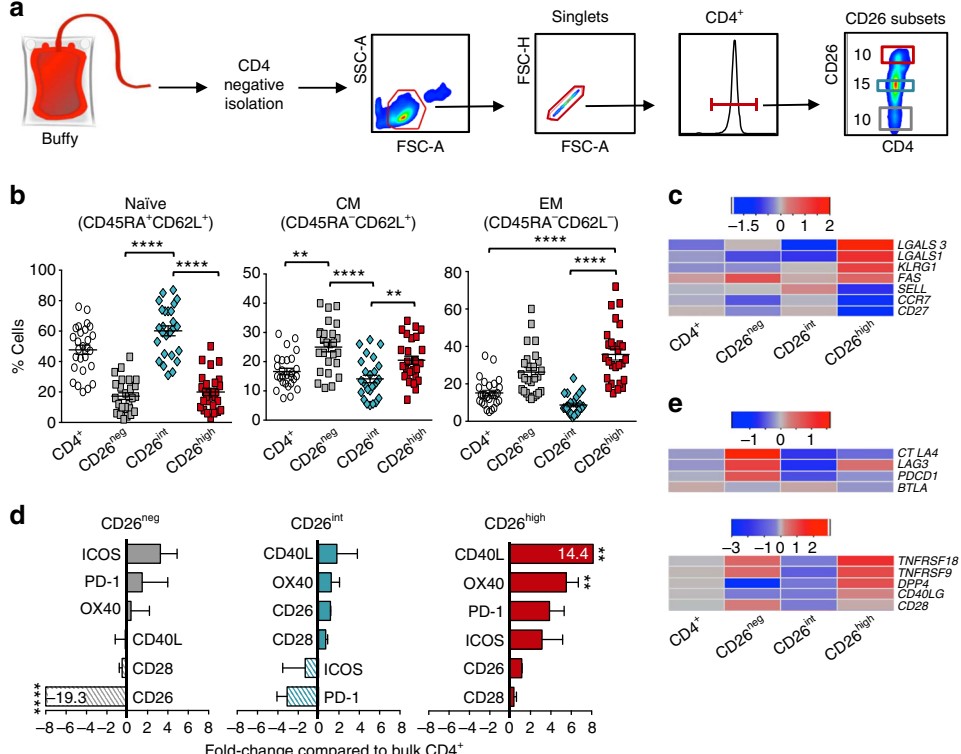

**Fig. 2** CD26$^{int}$ T cells are naive, whereas CD26$^{neg}$/CD26$^{high}$ T cells are differentiated. **a** Sorting strategy: CD4$^+$ T cells were isolated from buffy coats from healthy individuals and FACS-sorted into bulk CD4$^+$, CD26$^{neg}$ (bottom ~10%), CD26$^{int}$ (middle ~15%), and CD26$^{high}$ (top ~10%). **b, c** Memory phenotype for all subsets was determined using flow cytometry (**b** N = 26) and gene array analysis (**c** N = 3-5) prior to bead stimulation. For **c**, RNA was isolated and gene expression assessed by OneArray. Heat map displays (+/−) log2-fold change in memory-associated genes. **d, e** Graphical representation of co-stimulatory and co-inhibitory markers determined by flow cytometry (**d** N = 20-26) and gene array (**e** N = 3-5) prior to bead stimulation. Surface marker expression in **d** was calculated and graphed as a fold change of CD26$^{neg}$, CD26$^{int}$, and CD26$^{high}$ T cells compared to bulk CD4$^+$. P values were calculated using a One-way ANOVA with a Kruskal Wallis comparison. Error bars represent mean ± SEM. **P < 0.01; ****P < 0.0001

the tumor. Collectively, our findings provide new insight into CD26 for the advancement of T cell-based cancer immunotherapies in the clinic.

## Results

**CD26$^{high}$ T cells are activated and regress established tumors.** CD26 is expressed on effector and memory, but not regulatory (Tregs), CD4$^+$ T cells[27, 28]. Yet, it remains unknown whether CD26 correlates with these opposing subsets in cancer therapy. To address this question, we flow-sorted murine TRP-1 CD4$^+$ T cells, which express a transgenic TCR specific for tyrosinase on melanoma, via CD26 expression. This strategy enriched CD4$^+$ T cells into two groups: CD26$^{neg}$ and CD26$^{high}$. Strikingly, a mere 50,000 CD26$^{high}$ T cells were more effective at clearing B16F10 melanoma tumor than 50,000 CD26$^{neg}$ T cells when infused into lymphodepleted mice (Supplementary Fig. 1a, b). Moreover, half of the mice treated with CD26$^{high}$ T cells experienced a curative response (Supplementary Fig. 1c).

We next determined the antitumor activity of human CD26$^{high}$ T cells in a CAR-T model. To do this, we used the strategy depicted in Fig. 1a. First, CD4$^+$ T cells were isolated from the peripheral blood of a healthy individual and sorted by CD26 expression. Following bead activation, CD26$^{neg}$ and CD26$^{high}$ T cells were transduced to express a chimeric antigen receptor that targets mesothelin and signals CD3ζ (MesoCAR; ~98% CAR-specific; Supplementary Fig. 1d). Similar to murine cells, human CD26$^{high}$ T cells ablated large, human mesothelioma

tumors in NSG mice to a greater extent than CD26$^{neg}$ T cells (Fig. 1b). Our findings show that both murine and human T cells that express high levels of CD26 can effectively regress solid tumors in vivo.

As FoxP3$^+$ Treg cells express nominal CD26[28], we suspected that CD26$^{high}$ T cells exhibited greater antitumor activity because they were not suppressive (i.e., FoxP3$^{−/low}$). Surprisingly, CD26$^{high}$ T cells express more FoxP3 than CD26$^{neg}$ T cells (Fig. 1c). However, FoxP3$^+$CD26$^{high}$ T cells did not co-express Helios, a transcription factor expressed on thymus-derived Tregs (Fig. 1d, e; Supplementary Fig. 1e). In confirmation of these findings, very few CD4$^+$FoxP3$^+$Helios$^-$ T cells found in the bulk CD4$^+$ population lacked CD26 expression (~19%), whereas ~88% of CD4$^+$FoxP3$^+$Helios$^+$ Tregs were CD26 negative (Supplementary Fig. 1f). Given that human T cells upregulate FoxP3 following activation[29], we posited that CD26$^{high}$ T cells express FoxP3 because they exist in an activated state post-enrichment, but prior to ex vivo bead activation. As expected, CD26$^{high}$ T cells had a ten-fold higher expression of the activation markers CD25 and CD69 (~20%) than CD26$^{neg}$ T cells (~2%) (Supplementary Fig. 1g, h). CD26$^{neg}$ T cells also displayed a higher frequency of CD25$^+$CD127$^-$ cells (CD26$^{neg}$ ~12%; CD26$^{high}$ ~5%), an extracellular phenotype associated with Tregs[30, 31] (Fig. 1f). Finally, we discovered that ~74% of FoxP3$^+$CD26$^{high}$ T cells secreted the cytokines IL-2 and/or IFNγ (Fig. 1g). Conversely, <9% of CD26$^{neg}$ T cells were capable of cytokine production. Collectively, our data reveal that a portion of CD26$^{high}$ T cells are activated post-enrichment and clear tumors when redirected with a CAR.

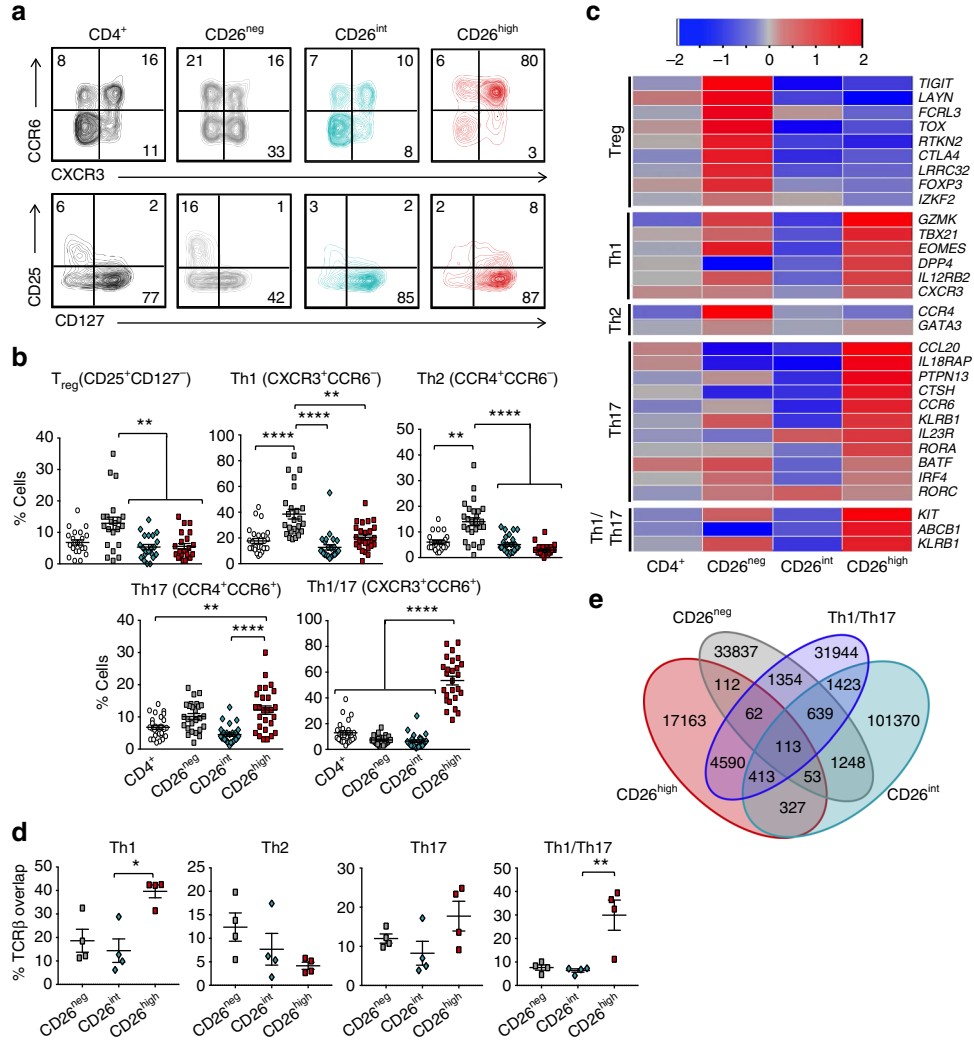

**Fig. 3** CD26 defines CD4+ T cells with naive, helper, or regulatory properties. CD4+ T cells were isolated from healthy donors and sorted by CD26 expression (Fig. 2a). Representative FACS plots (**a**) and phenotype data (**b** N = 26) from multiple healthy individuals. **c** Heat map of (+/−) log2-fold change in expression of CD4+ subset-associated genes (N = 3–5) prior to bead stimulation. **d, e** Human CD4+ T cells were sorted into bulk CD4+, CD26neg, CD26int, CD26high, Th1 (CXCR3+CCR6−), Th2 (CCR4+CCR6−), Th17 (CCR6+CCR4+), and Th1/Th17 (CXCR3+CCR6+). DNA was isolated from sorted T cells prior to expansion of TCRβ sequences using an immunoSEQ kit and subsequent analysis. Data shown is the percent TCRβ overlap between groups (**d**) and a Venn diagram (**e**) displaying the overlap frequencies of identical TCRβ sequences between CD26neg, CD26int, CD26high, and Th1/Th17 subsets (N = 4). P values for **b** and **d** were calculated using one-way ANOVA with a Kruskal-Wallis comparison. Error bars represent mean ± SEM. *P < 0.05; **P < 0.01; ****P < 0.0001

**Naive and memory CD4+ T cells have variant CD26 expression.** Although human CD26high T cells are more effective at clearing tumor than CD26neg cells, these cells are differentiated and might not be ideal for ACT. As CD26int cells have briefly been described as naive (CCR7+CD45RA+)[24], we hypothesized that they would elicit superior antitumor immunity compared to CD26neg or CD26high T cells. To test this, we first performed a detailed characterization of their phenotype in vitro. CD4+ T cells were enriched from healthy donors and FACS-sorted into bulk CD4+, CD26neg, CD26int or CD26high subsets (sorting purity >80%—Fig. 2a; Supplementary Fig. 2a). As expected, ~65% of CD26int cells were naive (CD45RA+CD62L+), whereas more than half of CD26neg and CD26high T cells possessed a more differentiated central or effector memory phenotype (Fig. 2b; Supplementary Fig. 2b). Furthermore, we found a heightened expression of CD95 on CD26high T cells (Supplementary Fig. 2c). Gene array analysis corroborated our findings, showing both CD26neg and CD26high T cells expressed less CCR7, CD27 and CD62L than CD26int

T cells (Fig. 2c). Furthermore, CD26high cells expressed heightened levels of markers associated with effector memory T cells (LGALS1, LGALS3)[32] as well as the senescence marker KLRG1 and FAS death ligand. Importantly, the differentiated phenotype of CD26neg and CD26high T cells and naive characteristics of CD26int T cells were reproducibly observed in more than 25 healthy donors.

Given that CD26int cells are naive, we posited that they would express less co-stimulatory and co-inhibitory receptors than CD26neg and CD26high T cells. As expected, CD26int T cells expressed less co-stimulatory and co-inhibitory markers than CD26high T cells (Fig. 2d; Supplementary Fig. 2d). Conversely, CD26high T cells expressed more PD-1, CD40L, and OX40 than the other subsets, whereas ICOS and PD-1 were the most prevalent markers on CD26neg. Furthermore, gene array analysis revealed an upregulation of CTLA4,LAG3, and PDCD1 in both CD26neg and CD26high T cells compared to CD26int (Fig. 2e, top). Despite the increased co-inhibitory markers on CD26high T cells,

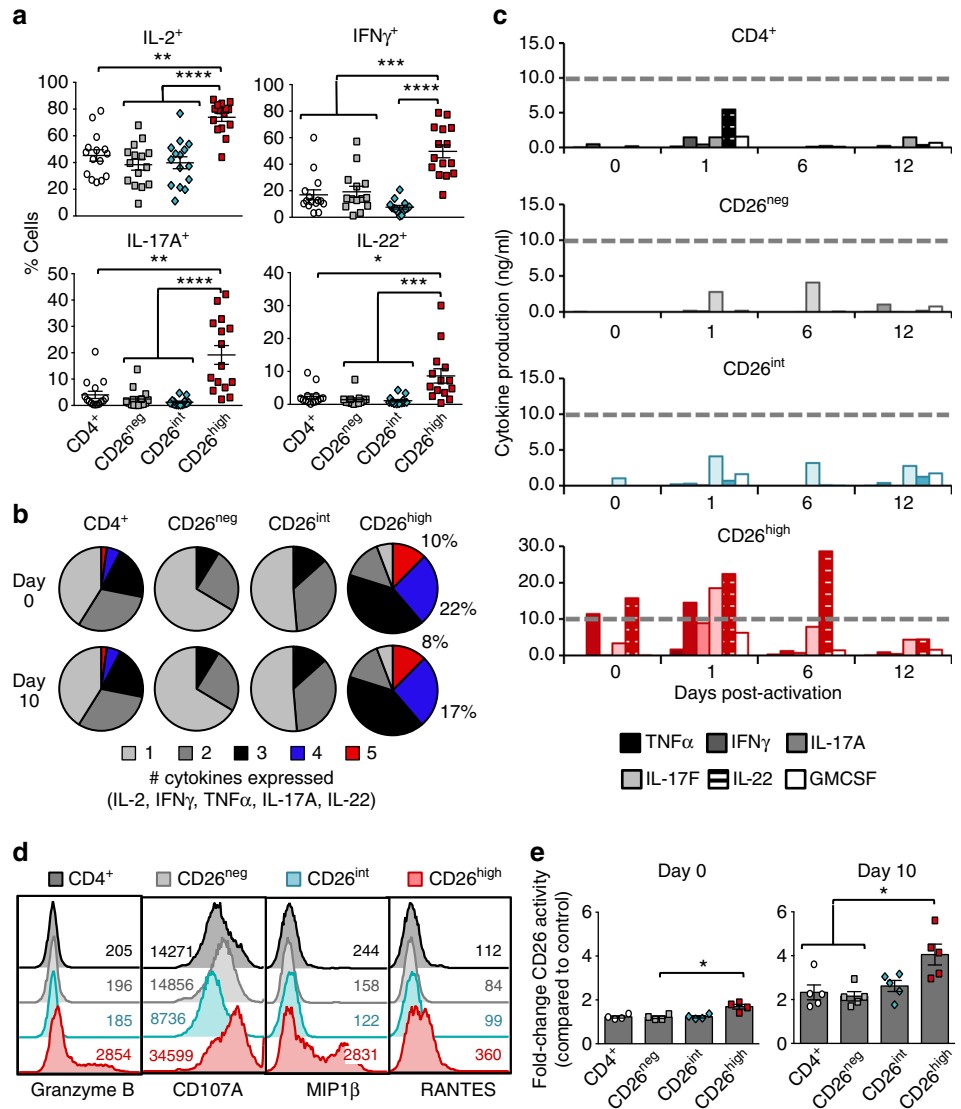

**Fig. 4** CD26[high] T cells are multi-functional and enzymatically active. CD4[+] T cells from healthy individuals were isolated and sorted by CD26 expression (Fig. 2a). **a**, **b** Sorted T cells were activated with PMA/Ionomycin and Monensin for 4 h prior to intracellular staining (N = 26). In **b**, three independent donors were analyzed by FlowJo software and graphed to display the percentage of cells simultaneously secreting 1–5 cytokines (IL-2, IFNγ, TNFα, IL-17A, IL-22). **c** Supernatant was collected from cells at pre-activation (0) and 1, 6, and 12 days post-activation time points for ELISA (N = 2; data shown is representative of two individual experiments). **d** T cells activated in vitro were subjected to intracellular staining (N = 5–11). **e** 1e[5] sorted cells per group from pre-activation (0) and 10 days post-activation were cultured with the CD26 ligand gly-pro-P-nitroanalide for 2 h at 37 °C and analyzed for colorimetric changes to determine enzymatic activity (N = 5). P values in **a** and **e** were calculated by One-way ANOVA with a Kruskal-Wallis comparison. Error bars represent mean ± SEM. *P < 0.05; **P < 0.01; ***P < 0.001; ****P < 0.0001

they also expressed many co-stimulatory markers, including CD40L (*CD40LG*), 41BB (*TNFRSF9*), and GITR (*TNFRSF18*) (Fig. 2e, bottom). Overall, these findings reveal that CD26[int] T cells possess a naive phenotype and express less co-stimulatory/inhibitory receptors compared to CD26[neg] and CD26[high] T cells.

**CD26 expression correlates with specific CD4[+] T cell subsets.** Although CD26[neg] T cells contain a Treg population[28] and CD26[high] cells have been published to exhibit a Th1[33] or Th17[24] phenotype, CD26[int] cells have not yet been characterized. Given that CD26[int] T cells appear naive, we hypothesized that they would not express the chemokine receptors indicative of any particular subset, whereas CD26[neg] and CD26[high] T cells would exhibit their reported Treg and Th1/Th17 phenotypes. As expected, CD26[high] T cells were composed of Th1 (CXCR3

[+]CCR6[−]) and Th17 (CCR4[+]CCR6[+]) cells, with the majority being hybrid Th1/Th17 (CXCR3[+]CCR6[+]) cells (Fig. 3a, b). These cells also exhibited heightened levels of CD161, which has recently been identified as a marker for long-lived antigen-specific memory T cells[34] (Supplementary Fig. 3a). We also identified Th1, Treg (CD25[+]CD127[−]), and Th2 (CCR4[+]CCR6[−]) populations in the CD26[neg] subset. On the contrary and as expected, CD26[int] cultures were comprised of lower frequencies of helper memory subsets. As shown in Fig. 3c, gene array analysis confirmed that CD26[neg] cells expressed a Treg (i.e., *TIGIT*, *TOX*, *CTLA4*), Th1 (i.e., *EOMES*, *GZMK*, *IL12RB2*) and Th2 (*CCR4*, *GATA3*) signature. Conversely, CD26[high] T cells had high gene expression in the Th1 (i.e., *GZMK*, *TBX21*, *EOMES*), Th17 (i.e., *CCL20*, *IL18RAP*, *PTPN13*), and Th1/Th17 (*KIT*, *ABCB1*, *KLRB1*) subsets. As expected, the subset-defining genes in CD26[int] T cells were sparse. Collectively, these findings confirm

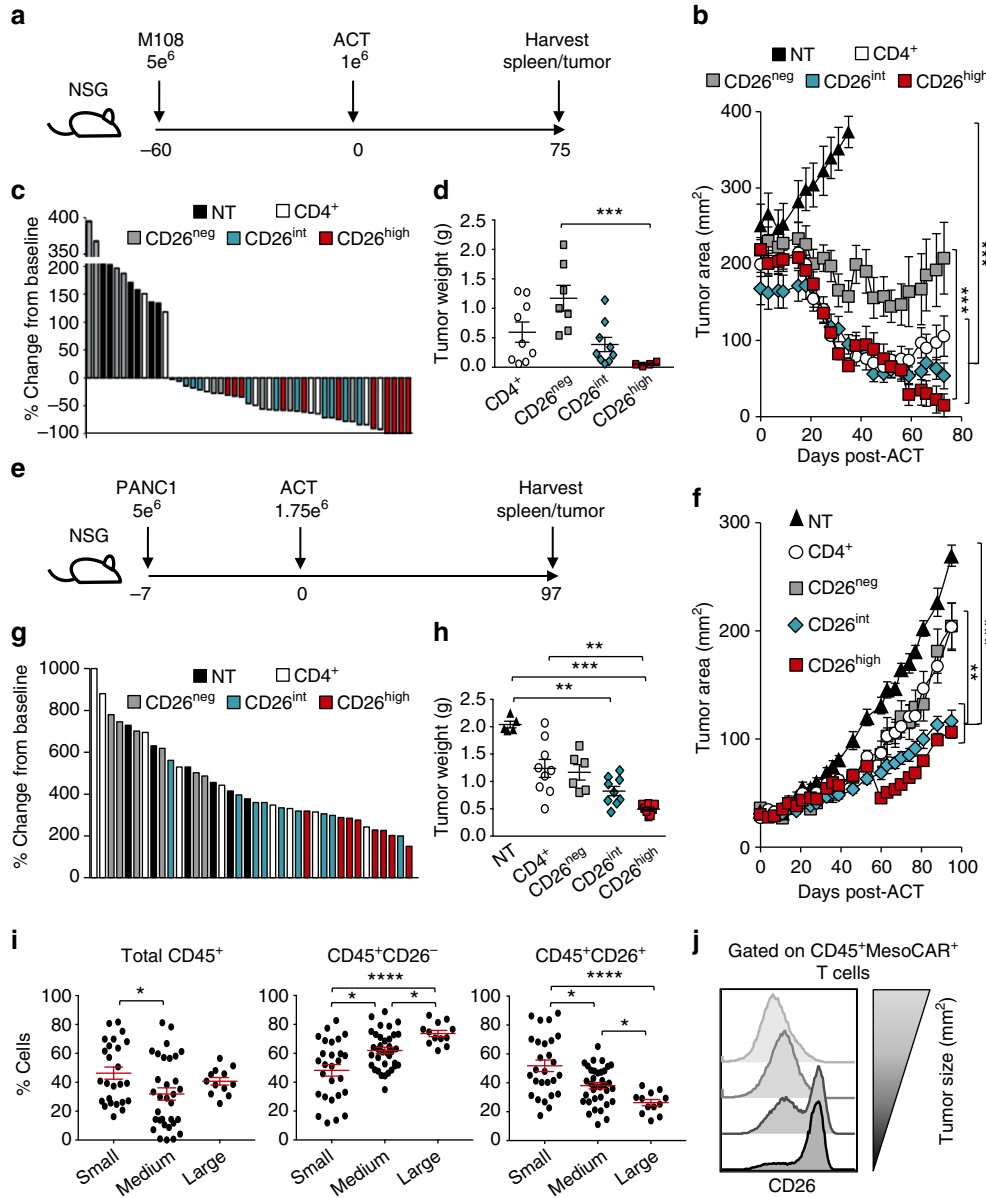

**Fig. 5** Human CD26int and CD26high T cells regress/slow established tumors. CD4+ T cells were isolated from healthy individuals, sorted by CD26 expression, transduced to express MesoCAR, and expanded for 10 days. **a, b** NSG mice bearing large M108 mesothelioma tumor (established for 60 days) were infused with 1e6 human CD4+, CD26neg, CD26int, or CD26high T cells. Post-ACT, tumors were measured bi-weekly until mice were killed and organs harvested at 75 days post-ACT (N = 7–9 mice per group). P values for the tumor curve were calculated by one-way ANOVA with a Kruskal-Wallis comparison on the final day when mice from all comparison groups were still alive (NT vs. all groups = day 38; CD26neg vs. CD26high = day 59). **c** Percent change in tumor size from baseline (day 0) to endpoint (day 75) was calculated and graphed as a Waterfall plot. **d** Graphical representation of tumor weights (g) harvested from treated mice 75 days post-ACT. **e, f** NSG mice bearing established pancreatic tumors (PANC1) were infused with 1.75e6 human CD4+, CD26neg, CD26int, and CD26high T cells and tumors were measured bi-weekly for more than 3 months (N = 6–9 mice per group). P values for the tumor curve were calculated by one-way ANOVA with a Kruskal-Wallis comparison on the final day when mice from all comparison groups were still alive (All groups = day 84). **g** Graphical representation of tumor weight (g) harvested from mice 97 days post-ACT. **h–j** Tumors from all treated mice were harvested, digested, and run through a strainer. Resulting cell suspension was stained for flow cytometry and graphed relative to tumor size (small <100 mm², medium = 100–200 mm², large >200 mm²). P values for **d**, **h** and **i** were calculated by one-way ANOVA with a Kruskal-Wallis component. Data with error bars represent mean ± SEM. *P < 0.05; **P < 0.01; ***P < 0.001; ****P < 0.0001

that CD26int T cells are naive and not yet committed to any subsets.

Helper T cell subsets express a specific T cell receptor β (TCRβ) profile[35]. Using TCRβ analysis, we found that many TCRs in CD26high T cells overlap with Th1, Th17, and Th1/Th17 cells, but not with Th2 cells (Fig. 3d). Although CD26high T cells had greater overlap with Th1 cells than CD26neg did, the overlapping TCRs between Th1 and CD26neg were more highly expressed

(Supplementary Fig. 3b). This heightened TCRβ expression revealed a correlation between CD26neg and Th1 cells (r² = 0.297), as well as CD26high and Th1/Th17 cells (r² = 0.526; Supplementary Fig. 3c), thereby confirming our previous findings. Although CD26high T cells exhibited significant overlap with Th1/Th17 cells, they shared little TCRβ overlap with CD26neg or CD26int cells (Fig. 3e; Supplementary Fig. 3d, e). Likewise, CD26int cells had minimal TCRβ overlap with any of

the Th subsets. Overall, these data confirm that CD26$^{high}$ and CD26$^{neg}$ T cells are differentiated cells containing multiple cell subsets while CD26$^{int}$ T cells are primarily uncommitted.

**CD26$^{high}$ cells are multi-functional and enzymatically active**. Given that CD26$^{neg}$ and CD26$^{high}$ T cells are differentiated, we hypothesized that they would secrete more cytokines than naive CD26$^{int}$ in vitro. Interestingly, CD26$^{high}$ T cells had heightened production of multiple cytokines, including IL-2, IFNγ, IL-17A, IL-22, and TNFα (Fig. 4a; Supplementary Fig. 4a, b). However, this heightened cytokine profile did not directly correlate with cell differentiation, as evidenced by similar cytokine production between bulk CD4$^+$, CD26$^{neg}$, and CD26$^{int}$ T cells. Given the numerous cytokines produced by CD26$^{high}$ T cells, we next investigated their capacity to secrete multiple cytokines at once (i.e., multi-functionality). Strikingly, CD26$^{high}$ T cells were highly multi-functional (Supplementary Fig. 4c) with roughly 25% of these cells producing 4–5 cytokines simultaneously (Fig. 4b), a phenomenon not seen in the other subsets. When we took into account the percentage of these donors not producing any cytokines, we discovered that 60–80% of CD4$^+$, CD26$^{neg}$, and CD26$^{int}$ T cells were incapable of producing cytokines on day 0, whereas the majority of CD26$^{high}$ T cells produced 3–5 cytokines simultaneously (Supplementary Fig. 4d). This vast cytokine production was maintained in CD26$^{high}$ T cells throughout culture (Fig. 4c). Furthermore, CD26$^{high}$ T cells produced more cytotoxic granules (Granzyme B, CD107A) and chemokines (MIP1β, RANTES) compared to other subsets, a property normally used to define cytotoxic CD8$^+$ T cells (Fig. 4d; Supplementary Fig. 4e).

As CD26 enzymatic activity has a minor role in supporting T cell function[25], we next posited that enzymatic activity would be elevated in CD26$^{high}$ T cells compared to the other subsets. Indeed, throughout a 10-day expansion in vitro, CD26$^{high}$ T cells maintained the highest enzymatic activity (Fig. 4e; Supplementary Fig. 4f). Thus, CD26$^{high}$ T cells are uniquely multi-functional, cytotoxic, and enzymatically active.

**CD26 subsets exist in peripheral blood of cancer patients**. Although CD4$^+$ T cells with distinct CD26 expression profiles exhibit unique immunological properties in healthy donors, it is unclear whether these biological assets are similar in cancer patients. To address this question, we isolated T cells with high, intermediate, or low CD26 expression from the blood of patients with malignant melanoma and examined their function, phenotype, and memory profile. CD26 was distributed similarly on CD4$^+$ T cells from melanoma patients (Supplementary Fig. 5a) as in normal donors (Fig. 2a). Furthermore, the mean fluorescence intensity (MFI) of CD26 on all subsets was similar between cancer patients and healthy individuals, with CD26$^{high}$ T cells maintaining the greatest CD26 expression (Supplementary Fig. 5b). The phenotype of CD26$^{neg}$, CD26$^{int}$, and CD26$^{high}$ T cells in melanoma patients was similar to that seen in health donors. For example, CD26$^{int}$ T cells were naive, denoted by high CD45RA, CD62L, and CCR7 markers (Supplementary Fig. 5c). CD26$^{high}$ T cells expressed Th1/Th17 chemokine receptors (CCR6, CXCR3, and CD161), whereas CD26$^{neg}$ T cells expressed CXCR3, confirming that cell subsets within these cultures are comparable between cancer patients and healthy individuals (Supplementary Fig. 5d). Finally, CD26$^{neg}$ T cells from cancer patients contained Tregs, as they expressed less CD127 but greater CD39, FoxP3, and Helios (Supplementary Fig. 5e–f). CD26$^{high}$ T cells also secreted more IL-2, TNFα, IL-17A, IFNγ, and MIP-1β than other subsets following stimulation with PMA and Ionomycin (Supplementary Fig. 5g, h). CD26$^{neg}$ T cells from melanoma patients secreted elevated IL-4 (Supplementary Fig. 5i),

correlating with our previous finding that Th2 cells are prevalent in this culture (Fig. 3). Finally, CD26$^{high}$ T cells expressed more CCR2 and CCR8 than other subsets (Supplementary Fig. 5j). As CCR8 promotes T cell trafficking to the skin, it is intriguing that this chemokine receptor is abundant on CD26$^{high}$ T cells from melanoma patients. Collectively, we found that CD26$^{neg}$, CD26$^{int}$, and CD26$^{high}$ T cells from melanoma patients possess a similar biological profile as those identified in healthy donors.

**CD26$^{int}$ and CD26$^{high}$ T cells regress/inhibit tumor growth**. As enzymatically active CD26$^{high}$ T cells are more differentiated than CD26$^{int}$ T cells, we hypothesized that CAR$^+$CD26$^{int}$ T cells would clear tumor and persist better than CAR$^+$CD26$^{high}$ T cells. To address this hypothesis, we redirected human CD4$^+$, CD26$^{neg}$, CD26$^{int}$, and CD26$^{high}$ T cells to recognize mesothelin via a MesoCAR (Supplementary Fig. 6a). Ten days post-expansion, CAR-T cells were infused into NSG mice bearing large M108 mesothelioma (Fig. 5a). Surprisingly, despite the seemingly exhausted phenotype of CD26$^{high}$ T cells, they regressed tumor slightly, but not significantly, better than CD26$^{int}$ T cells (Fig. 5b; Supplementary Fig. 6b). CD4$^+$ T cells were only slightly less effective than CD26$^{int}$ or CD26$^{high}$, whereas 26$^{neg}$ T cells proved to be incapable of regressing mesothelioma. Our tumor curve data in Fig. 5b directly correlated with both percent tumor change from baseline (Fig. 5c) and tumor weights (Fig. 5d). Collectively, we found that CAR$^+$ T cells that express CD26 are more effective at clearing tumors.

In contrast to human mesothelioma mouse models, CAR-T therapy does not inhibit the growth of pancreatic tumors (Supplementary Fig. 6c). Thus, we next sought to determine if CAR-T cells that express CD26 (CD26$^{high}$ and CD26$^{int}$) could better control pancreatic cancer (PANC1) in mice to a greater extent than CD26$^{neg}$ T cells. To test this, we used a similar treatment strategy (Fig. 5e) as performed in our M108 model. Following CAR transduction (Supplementary Fig. 6d), both CD26$^{int}$ and CD26$^{high}$ T cells significantly slowed the progression of pancreatic tumors, whereas bulk CD4$^+$ and CD26$^{neg}$ T cells yielded little-to-no antitumor response (Fig. 5f; Supplementary Fig. 6e). These findings were confirmed by decreased tumor growth and weight in mice treated with CD26$^{int}$ or CD26$^{high}$ T cells compared to mice treated with CD4$^+$ or CD26$^{neg}$ T cells (Fig. 5g, h). Overall, our data demonstrate that differentiated CD26$^{high}$ T cells exhibit a similar antitumor response as naive CD26$^{int}$ T cells.

Given that CD26$^{int}$ and CD26$^{high}$ T cells yielded the best antitumor response, we next posited that CD26 expression on T cells in the tumor itself might correlate with overall response. To test this, we isolated tumor-infiltrating lymphocytes from all mice at experimental endpoint and assessed donor cell persistence and phenotype by flow cytometry. When graphed by tumor size (small <100 mm$^2$, medium = 100–200 mm$^2$, large >200 mm$^2$), we unexpectedly discovered that the percentage of CD45$^+$ T cells was similar between the groups and had no correlation with antitumor response (Fig. 5i, left). On the contrary, we found that CD26 expression on TIL tightly correlated with response as smaller tumors exhibited heightened percentages of CD26$^+$ TIL (Fig. 5i, middle and right; Fig. 5j).

We next sought to determine whether CD26 expression on the endogenous TIL of mice that had not received therapy also correlated with tumor size. To do this, mice were subcutaneously injected with B16F10 melanoma and CD26 expression on TIL was assessed by flow cytometry. Similar to our ACT model, we found that the expression of CD26 on endogenous TIL decreased as tumor size increased (Supplementary Fig. 6f). These observations not only exhibit that distinct CD26−expressing CD4$^+$ T cell

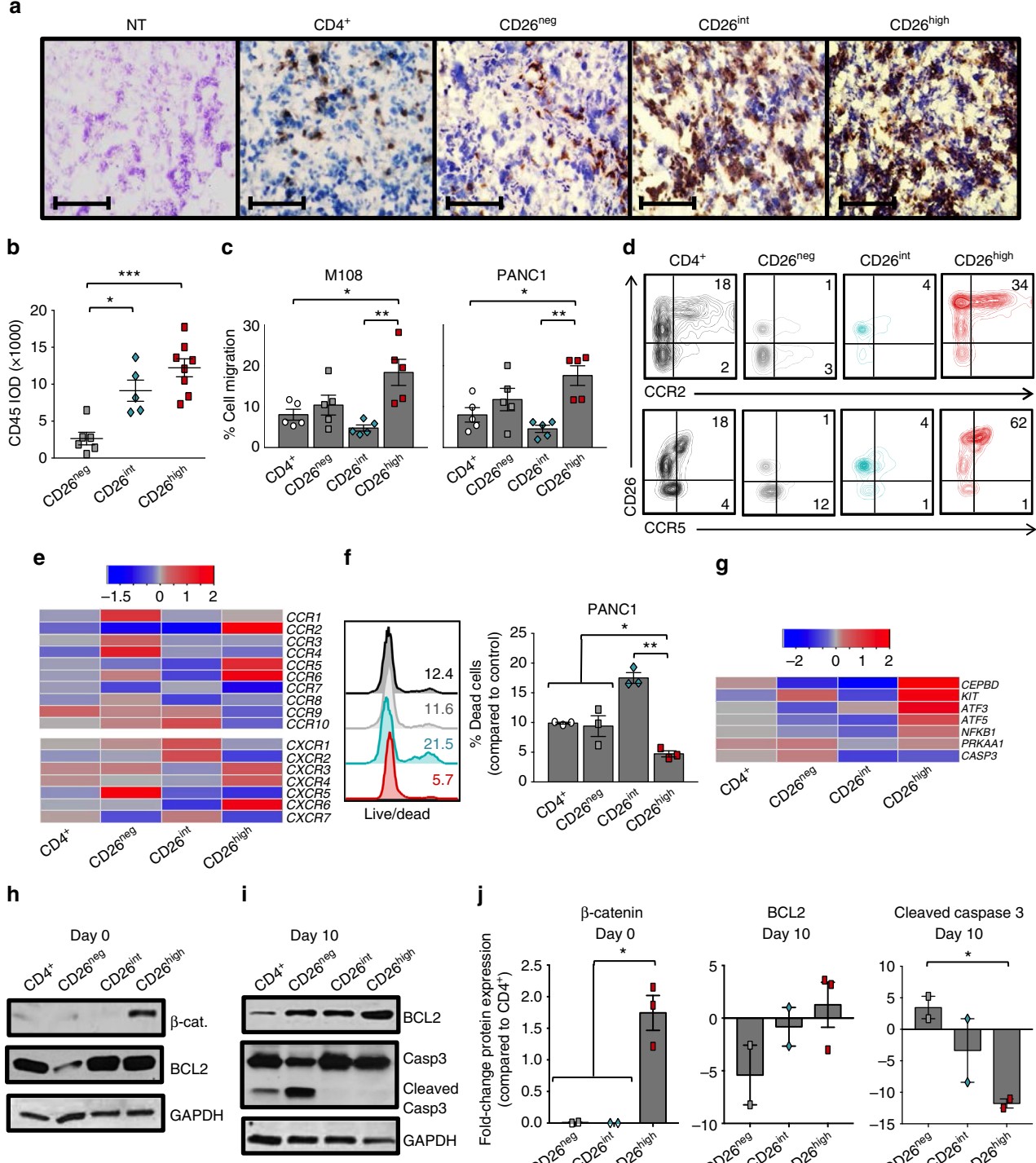

**Fig. 6** CD26[high] T cells have stemness and increased migratory capacity. **a**, **b** Tumors from treated PANC1-bearing mice (from Fig. 5) were harvested and then frozen in cryomatrix. Tumor samples were then sliced and used for immunohistochemistry analysis (purple = H&E, brown = CD45; $N$ = 5–9 tumors per group). Magnification = ×10. In **b**, the integrated optical density (IOD) of CD45 in CD26-sorted groups was quantified with ImageJ software and graphed ($N$ = 5–8 per group). **c** Sorted T cells were activated with CD3/ICOS beads and expanded in 100 IU/ml IL-2 for 10 days prior to testing cell migration via a transwell assay. $0.75e^6$ sorted cells were re-suspended and assessed for percent cell migration towards M108 or PANC1 supernatant in a 2 h time period ($N$ = 5). **d**, **e** Sorted T cells were analyzed for chemokine receptor expression by flow cytometry (**d** $N$ = 12–15) and gene array (**e** $N$ = 3–5) prior to bead stimulation. In **e**, data shown in heat map as (+/−) log2-fold change in chemokine receptor expression compared to bulk CD4[+]. **f** Viability of T cells that migrated towards PANC1 was determined by live/dead staining ($N$ = 3). **g** Anti-apoptotic and stemness genes were analyzed and displayed as a (+/−) log2-fold change compared to bulk CD4[+] ($N$ = 3–5). **h**–**j** Protein from pre-activation (day 0) and post-activation (day 10) cells was isolated and used for western blot analysis ($N$ = 2–3). In **j**, the fold change of β-catenin, BCL2 and cleaved Caspase-3 for each subset compared to bulk CD4[+] T cells was graphed ($N$ = 2–3). $P$ values were calculated using a One-way ANOVA with a Kruskal-Wallis comparison. Data with error bars represent mean ± SEM. *$P$ < 0.05; **$P$ < 0.01; ***$P$ < 0.001

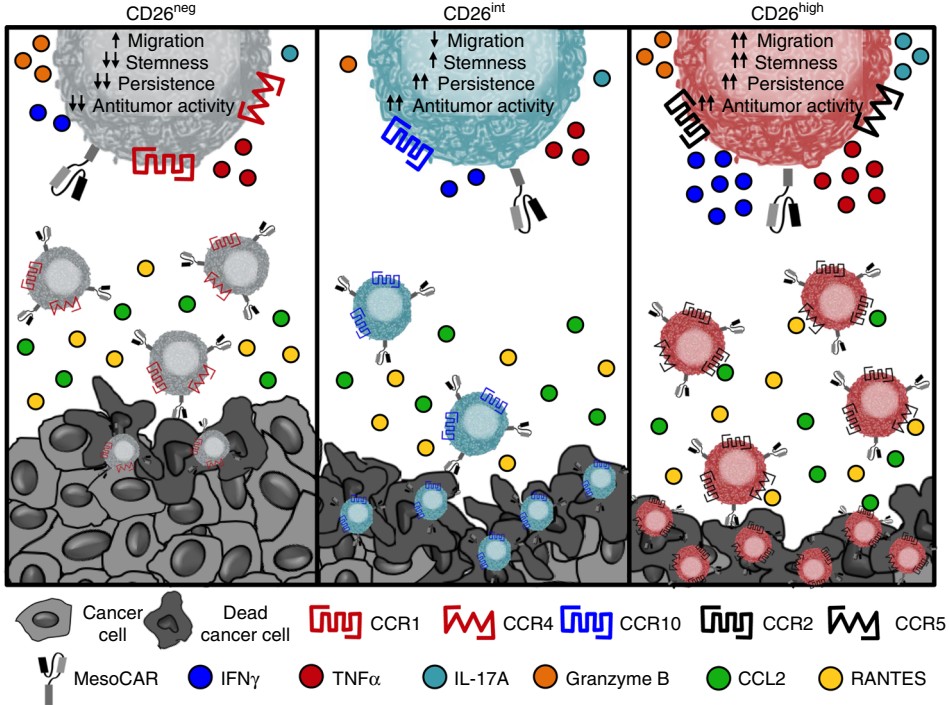

**Fig. 7** CD26 identifies three CD4$^+$ T cell subsets with distinct properties. Depiction of our observations on CD26 expression and cellular therapy described herein. CD26$^{neg}$ T cells, despite their enhanced capacity to migrate, fail to regress tumors due to regulatory properties, decreased persistence, and increased sensitivity to cell death. CD26$^{int}$ and CD26$^{high}$ T cells exhibit similar antitumor activity, but have vastly different immunological properties. Despite their decreased migration, CD26$^{int}$ T cells are naive and capable of persisting long-term. CD26$^{high}$ T cells, despite their differentiated phenotype, exhibit several anti-apoptotic and stemness features, persist long-term, co-secrete multiple cytokines, and cytotoxic molecules and have a natural capacity to migrate towards various established solid tumors

subsets exist in both endogenous and adoptively transferred TIL, but also identify a direct correlation between CD26 and their subsequent ability to slow/regress tumors.

Although we found that CD4$^+$CD26$^{high}$ T cells exhibit greater antitumor activity than other CD4$^+$ subsets, how they compare to the commonly used cytotoxic CD8$^+$ T cells is unknown. To test this concept, human CD4$^+$CD26$^{high}$ and CD8$^+$ T cells were isolated and engineered to be MesoCAR$^+$. An in vitro cytotoxic assay using mesothelin-expressing K562 cells revealed that CD4$^+$CD26$^{high}$ T cells have a similar, if not enhanced, capacity to kill cancer compared to CD8$^+$ cells (Supplementary Fig. 6g). Furthermore, adoptive transfer into mesothelioma-bearing mice showed that CD4$^+$CD26$^{high}$ T cells clear tumors significantly better than traditionally used CD8$^+$ T cells (Supplementary Fig. 6h, i). This finding, in addition to our previously discussed NSG mouse models in which no inherent immune system exists, reveals that CD4$^+$CD26$^{high}$ T cells are directly cytotoxic and capable of clearing tumors in the absence of other immune cells, including CD8$^+$ T cells.

**CD26$^{high}$ T cells have enhanced migration and stemness.** We next sought to determine the mechanisms underlying the effectiveness of CD26$^{int}$ and CD26$^{high}$ T cells. Owing to the multifunctional and cytotoxic nature of CD26$^{high}$ T cells (Fig. 4; Supplementary Fig. 4), we hypothesized that they would cause immediate tumor regression in mice, but would not persist like CD26$^{int}$ lymphocytes. Surprisingly, CD26$^{high}$ T cells persisted as well as CD26$^{int}$ cells in the tumor (Supplementary Fig. 7a). Conversely, few CD26$^{neg}$ T cells were detected. These findings were confirmed by immunohistochemistry, which revealed a direct correlation between CD26 and enhanced CD45$^+$ donor cell

persistence in the tumor (Fig. 6a, b). Collectively, these findings reveal that CD26$^{int}$ and CD26$^{high}$ T cells persist in the tumor to a greater extent than bulk CD4$^+$ and CD26$^{neg}$ T cells.

Given the ample amount of donor T cells persisting in the tumors of mice treated with CD26$^{int}$ and CD26$^{high}$ T cells, we next sought to determine if they have an enhanced capacity to migrate. To test this, sorted human CD4$^+$ T cells were activated and expanded in culture for 10 days as previously described. On day 10, cells from each group were placed in the top well of a transwell plate and assayed for their migration towards tumor-secreted supernatant (or controls) following a 2 h incubation period. CD26$^{neg}$ and CD26$^{high}$ T cells migrated slightly better in control conditions than bulk CD4$^+$ and CD26$^{int}$ T cells (Supplementary Fig. 7b), perhaps due to the elevated expression of genes that aid in migration, including *RGS1, RHOA, ROCK1,* and *DIAPH1* (Supplementary Fig. 7c). Furthermore, we discovered that CD26$^{high}$ T cells migrated to M108 and PANC1 in vitro to a greater extent than bulk CD4$^+$, CD26$^{neg}$, or CD26$^{int}$ T cells (Fig. 6c). Given that the chemokine receptors CCR2 and CCR5 have been shown to be important in migration towards mesothelioma tumors, we next observed the expression of these receptors on CD26$^{high}$ T cells. Indeed, we found that CD26$^{high}$ T cells not only expressed elevated levels of CCR2 and CCR5, but also expressed more CCR6, CXCR3, CXCR4, and CXCR6 (Fig. 6d, e; Supplementary Fig. 7d). Interestingly, the ligands for these chemokine receptors (CCL2, RANTES, CCL20, CXCL9-10, CXCL12, and CXCL16, respectively) are produced by pancreatic tumors[36]. In addition, CD26$^{high}$ T cells were more viable following migration (Fig. 6f).

Given this finding, we next tested if CD26$^{high}$ T cells were resistant to apoptosis. We found that CD26$^{neg}$ T cells were more apoptotic than bulk CD4$^+$, CD26$^{int}$, and CD26$^{high}$ T cells

(Supplementary Fig. 7e). Conversely, CD26[high] T cells expressed many anti-apoptotic genes, such as *KIT, CEBPD,* and *ATF3* (Fig. 6g). As Th17 cells have stemness and durable memory despite their differentiated appearance[22], we next assessed if CD26[high] T cells expressed proteins in the Wnt/β-catenin pathway. Compared to naive CD26[int] or regulatory CD26[neg] T cells, CD26[high] T cells had greater β-catenin expression (Fig. 6h) and maintained heightened BCL2 with minimal caspase-3 cleavage (Fig. 6i). These findings repeated among several healthy donors (Fig. 6j) and were confirmed by the upregulation of *Lef1* concurrently with CD26 expression (Supplementary Fig. 7f). Collectively, our findings reveal that CD26[high] T cells have remarkable self-renewal potential and a profound ability to migrate, survive, and persist in the tumor long-term.

As visualized in Fig. 7, CD26 identifies three human CD4[+] T cell subsets with distinct immunological properties. First, CD26[neg] T cells exhibit poor persistence and antitumor activity due to increased Treg expression and lack of stemness properties. CD26[int] T cells are mainly naive, persistent, and effectively slow/regress human tumors. Finally, CD26[high] T cells possess qualities of stemness and migration factors that support their persistence and antitumor activity in solid tumors.

## Discussion

The T cell subsets that optimally elicit robust memory responses to tumors have been hotly debated. Recent reports underscore the need for naive or less differentiated memory T cells to maintain a long-term antitumor response in vivo[37–39]. Unfortunately, T cells become unresponsive quickly during tumorigenesis[40]. Moreover, exhausted and terminally differentiated T cells are epigenetically stable, regardless of PD-1 blockade therapy, and thus fail to become memory T cells[41]. One advantage to ACT therapy is the ability to infuse selected T cells with long-lived immunity into cancer patients. Several markers, such as CD45RA, CCR7, CD62L, and CD95, identify naive or stem memory T cells[42]. Our data suggest that this long list of markers can be consolidated to one: CD26. Collectively, we demonstrate that CD26 expression identifies three distinct human T helper subsets—regulatory, naive, and stem memory—with varying levels of antitumor activity.

We discovered that CD26[int] T cells were naive, CD26[neg] cultures contained Treg, Th1 and Th2 cells, and CD26[high] T cells displayed a Th1/Th17 profile and were highly multi-functional/cytotoxic. Although cells with heightened cytotoxicity lyse tumors in vitro, they often have diminished replicative and antitumor capacity in vivo[39]. Surprisingly, CD26[high] T cells controlled pancreatic tumors and regressed mesothelioma slightly, but not significantly, better than CD26[int]. In addition to regressing tumors in multiple cancer models, CD26[high] T cells also exhibited long-term persistence. Similar to Th17 cells that persist in vivo via the Wnt/β-catenin signaling pathway despite their effector memory profile[22], we discovered that CD26[high] T cells display markers of stemness, such as increased β-catenin, BCL2 and Lef1. Studies similar to that by Graef et al.[21], which showed that a single antigen-specific mouse CD8[+]CD62L[+] T cell could reconstitute and protect mice from infection post-serial transfer, would be interesting to investigate in order to confirm the stemness of CD26[high] T cells. On the basis of our work herein, we posit that CD26[+] T cells would persist and protect the host from malignancy, whereas CD26[−] T cells would be short-lived and unable to prevent tumor growth. Also, CD26[high] T cells naturally expressed the chemokine receptors CCR2 and CCR5, which likely helped CD26[high] T cells migrate to the tumor site; a notion supported by data showing that transduction of CCR2 into transferred T cells

improved their trafficking to tumors[43, 44]. Thus, CD26[high] T cells employ key mechanisms, including enhanced stemness and migration, to persist and lyse tumor.

In addition, we found it intriguing that CD26[high] T cells express multiple co-stimulatory and co-inhibitory markers. Activating T cells via OX40/CD40L agonists[45, 46] or PD-1/LAG3 antagonists[1, 47, 48] enhances antitumor immunity. GITR, a co-stimulatory receptor known for it's expression on Tregs, has recently been shown to enhance effector T cell function and bestow protection against Treg-mediated immunosuppression[49, 50]. The elevated expression of these markers on CD26[high] T cells begs the question of whether checkpoint modulators regulate the frequency and function of these cells in cancer patients, thereby dictating treatment outcome. Therefore, the simple addition of a CD26 detection antibody in immune-monitoring cores in cancer centers could easily shed light on this key question.

Finally, we discovered that mice bearing melanoma, mesothelioma or pancreatic cancer experienced a better treatment outcome when a heightened percentage of CD26[+] donor T cells infiltrated the tumor. These data suggest a potential role of CD26 in tumor immunity, which could entail several mechanisms. First, our finding that human MesoCAR[+] T cells and endogenous mouse TIL expressing CD26 correlate with reduced tumor size suggests that CD26 augments antitumor immunity in an MHC-independent manner. Given the polyfunctionality of CD4[+]CD26[high] T cells (indicated by their ability to co-express IL-17A, IFNγ, IL-2, TNFα, and Granzyme B), our findings could be the result of enhanced inflammation at the tumor site (i.e., induction of a "hot" tumor), which has been shown to associate with tumor regression[51, 52]. It is also worth noting that human regulatory T cells express little-to-no CD26[28], a fact that could implicate that the enhanced regression seen in mice expressing a greater percentage of CD26[+] TIL is simply due to a greater Teff to Treg ratio at the tumor site. Finally, as CD26 increases on T cells following stimulation[53], its expression could also correlate with TIL activation in vivo.

Although the majority of our work herein utilized CAR-engineered CD4[+] T cells, we also identified a similar phenotypic and functional correlation between CD26 expression on both TCR-specific and endogenous TIL. Given that previous reports identify CD26 as a marker of memory T cells with an enhanced response to recall antigens[33], it is possible that the CD26[+] compartment of T helper cells express TCRs that recognize tumor antigens. In addition, recent reports have shown that mutation-specific CD4[+] T cells in a patient with metastatic cholangio-carcinoma have clinical benefit[54]. Intriguingly, the phenotype described by Tran et al., specifically the functional nature and increased OX40 and 4-1BB on erbb2 mutation-specific T helper cells, is similar to our CD26[high] T cells described herein. Therefore, our data might also imply that a small cohort of CD26-expressing CD4[+] T cells recognize mutated tumor antigens displayed on MHC II-positive tumors, which has been shown to mediate antitumor efficacy in patients treated with cell therapy and checkpoint modulators[54–59]. If this were true, CD26 could be an ideal marker to isolate CD4[+] TIL with responses against mutated antigens that can be exploited to regress established tumors. Collectively, our findings indicate that CD4[+]CD26[+] T cells could be used for TCR-based immunotherapy, as they may potentially mediate enhanced responses to mutated tumor antigen in an MHC II-restricted fashion. Thus, it will be worthwhile to deduce the TCR specificity on human CD4[+] TILs expressing low, intermediate, and high levels of CD26. Regardless of the putative mechanism by which CD26[+] T cells mount immunity to tumors, our data suggest that CD26 could have a role—be it functional or as a biomarker—in T cell-based therapies, including neoantigen-specific vaccines, ICB, and ACT.

One prevalent role of CD26 in autoimmune diseases such as diabetes is via the enzymatic activity, which inactivates peptides (GLP-1 and GIP) critical for stabilizing blood glucose levels[26]. Contrary to the known role of CD26 enzymatic activity in diabetes, its effect on antitumor activity has been controversial. One study found that CD26 inhibition following ACT or checkpoint therapy enhanced immunity by renewing the recruitment of CXCR3[+] T cells to the tumor[60]. On the contrary, inhibiting CD26 has also been shown to accelerate tumor metastasis via NRF2 activation[61]. Herein, we discovered that CD26[high] T cells maintain elevated enzymatic activity in vitro. Further exploration on the biological role of CD26 in antitumor immunity will be insightful. Genetic or pharmaceutical regulation of the enzymatic and multi-functional properties of CD26 will determine its role in T cell therapy and uncover if targeting CD26 can enhance immunotherapy in cancer patients.

To our knowledge, this is the first report that defines three human CD4[+] subsets by CD26 expression and examines their unique impact in tumor immunity. Data herein suggest that although CD26[int] and CD26[high] T cells are promising for cancer immunotherapy, CD26[neg] T cells are deleterious to this field but may have an important role in suppressing autoimmune diseases, such as GVHD or rheumatoid arthritis. Specifically, our major findings are threefold. First, we comprehensively characterized CD26[int] T cells and confirmed their naive phenotype. Given the extensive sorting or bead strategies required to isolate naive cells from patients, this finding could revolutionize ACT by simplifying the strategy for enriching ideal lymphocytes. Second, our findings reveal that CD26[high] T cells are not terminally differentiated, but rather have durable memory, enabling them to persist and regress multiple solid tumors. These discoveries appear to be, in part, due to their enhanced stemness properties and migration towards the tumor. Finally, our finding that mice with better responses had more CD26[+] T cells in both mesothelioma and pancreatic tumors reveals a possible significance of this marker in cancer immunotherapy.

## Methods

**Mice and tumor lines**. C57BL/6 (B6), TRP-1 TCR transgenic, and NOD scid gamma (NSG) mice were purchased from The Jackson Laboratory and housed in the comparative medicine department at the Medical University of South Carolina Hollings Cancer Center (MUSC, Charleston, SC). NSG mice were housed in microisolator cages to ensure specific pathogen-free conditions and given ad libitum access to autoclaved food and acidified water. All housing and experiments were conducted in accordance with MUSC's Institutional Animal Care and Use Committee's (IACUC) procedures and with the aid of the Division of Lab Animal Resources (DLAR). B16F10 (H-2b) melanoma (gift, N.P. Restifo), M108 mesothelioma (gift, C.H. June), and PANC1 pancreatic cancer (gift, M.R. Rubinstein) cells were utilized for tumor experiments.

**T cell subset isolation**. Peripheral blood cells from healthy, de-identified individuals were purchased as a buffy coat (Plasma Consultants) or a leukophoresis (Research Blood Components). Lymphocytes were enriched via centrifugation with Lymphocyte Separation Media (Mediatech). Untouched CD4[+] T cells were isolated by magnetic bead separation (Dynabeads, Invitrogen) and cultured overnight in CM and rhIL-2 (100 IU/ml; NIH repository). The next morning, CD4[+] T cells were stained with PE-CD26 (C5A5b; BioLegend) and v500-CD4 (RPAT4) or APCCy7-CD4 (OKT4; BDPharmingen) and sorted on a BD FACSAria Ilu Cell Sorter into bulk CD4[+], CD26[neg], CD26[int], and CD26[high]. Following sort, cells were cultured overnight in CM, 100 IU/ml rhIL-2, 10 µg/ml kanamycin/ampicillin, and 20 µg/ml anti-mycotic.

**T cell expansion**. *TRP-1*: Splenocytes from transgenic TRP-1 mice were isolated and cultured with 1 µl/ml TRP-1[106–130] peptide (SGHNCGTCRPGWRGAACNQKILTVR) and feeder T cells at a ratio of 1 feeder: 5 TRP-1 CD4[+] T cells. Cells were programmed to a Th17 phenotype with polarizing cytokines (10 ng/ml hIL-1β, 100 ng/ml hIL-21, 100 ng/ml hIL-6, 30 ng/ml hTGFβ, 10 µg/ml αm-IFNγ, 10 µg/ml αm-IL-4), and 100 IU/ml IL-2 for 6 days.

*Human*: Cells were cultured in CM supplemented with 100 IU/ml rhIL-2 at a 1:5 bead to T cell ratio using magnetic beads (Dynabeads, Life Technologies) decorated with antibodies to CD3 (OKT3) and ICOS (ISA-3, eBioscience), which

were produced in the lab according to manufacturer's protocol. Cells were de-beaded on day four and culture media/100 IU/ml rhIL-2 was replaced as needed.

**Flow cytometry**. Antibodies for extracellular stains were incubated with cells for 20 min in FACS buffer (PBS + 2%FBS). For intracellular staining, cells were activated with PMA/Ionomycin for 1 h, combined with Monensin (BioLegend) and incubated another 3 h prior to staining in Fix and Perm buffers (BioLegend). For transcription factors, the FOXP3 kit (BioLegend) was used according to manufacturer's protocol. Live/dead staining was performed using the Zombie Aqua Fixable Viability Kit (BioLegend). Data were acquired on a BD FACSVerse (BD Biosciences) and analyzed using FlowJo software (Tree Star). A complete list of antibodies can be found in Supplementary Table 1.

**MicroArray**. The Qiagen RNeasy Mini kit was used to isolate RNA from sorted CD4[+] T cells. Frozen RNA samples were sent to Phalanx Biotech Group for processing using their OneArray platform (San Diego, CA). NanoDrop ND-1000 was utilized to assess the quality and purity of the RNA. Absorbance ratios had a pass criteria of 260/280 ≥ 1.8 and A260/230 ≥ 1.5, indicating acceptable RNA purity. Agilent RNA 6000 Nano assay was used to ascertain RIN values, with a pass criteria of RIN value established at >6 indicating acceptable RNA integrity. Gel electrophoresis was used to evaluate gDNA contamination. Target preparation was performed using an Eberwine-based amplification method with Amino Allyl MessageAmp II aRNA Amplification Kit (AM1753, Ambion) to generate amino-allyl antisense RNA (aa-RNA). Prior to hybridization, labeled aRNA coupled with NHS-CyDye was prepared and purified. Purified coupled aRNA was quantified using NanoDrop ND-1000 with a pass criteria for CyDye incorporation efficiency at >15 dye molecular/1000 nt.

For data analysis, GPR files were loaded into Rosetta Resolver System. The Rosetta error model calculation was used to estimate random factors and systematic biases. Duplicate probes were averaged and median scaling was performed for normalization. Differentially expressed genes were defined as having a (+/−) log2-fold change ≥1 and $P < 0.05$. Where log2 ratios = "NA", the differences in intensity between the two samples had to be ≥1000.

Heatmaps were constructed in R (version 3.1.2) using gplots (version 2.16.0) log2 values for bulk CD4[+] T cells were averaged and used as a baseline for the genes of interest. For each sample, the fold change relative to baseline was calculated and the median value for the triplicates was used for generating figures.

**T cell receptor β sequencing**. CD4[+] T cells were isolated from four individual healthy individuals and sorted into the following groups: bulk CD4[+], CD26[neg], CD26[int], CD26[high], Th1 (CXCR3[+]CCR6[−]), Th2 (CCR4[+]CCR6[−]), Th17 (CCR4[+]CCR6[+]), and Th1/Th17 (CCR6[+]CXCR3[+]). Sorted subsets were then centrifuged and washed in PBS prior to extracting genomic DNA via Wizard Genomic DNA purification kit (Promega). Spectrophotometric analysis using NanoDrop (ThermoScientific) was used to assess the quantity and purity of genomic DNA. The ImmunoSEQ hsTCRβ kit (Adaptive Biotechnologies Corp, Seattle, WA) was used according to manufacturer's protocol to amplify the TCR genes. TCRβ sequencing was performed using the Illumina MiSeq platform at the Hollings Cancer Center Genomics Core. Analysis and graphing was performed using ImmunoSEQ software.

**ELISA**. T cells utilized for ELISA assays were plated at 2e[5] cells/200 µl CM for 12–15 h prior to supernatant collection. Supernatant was then used to detect IL-17A, IL-17F, IL-22, GMCSF, IFNγ, and TNFα by DuoSet ELISA kits (R&D) per manufacturer's instructions.

**CD26 enzymatic assay**. To determine CD26 enzymatic activity, a microplate-based fluorescence assay was performed. Briefly, 1e[5] T cells from days 0, 5, and 10 days post-activation were washed and re-suspended in 100 µl 4% gly-pro-P-nitroanalide in PBS. After 2 h at 37 ℃, the release of pNA from the substrate was assessed using a Multiskan FC plate reader (ThermoScientific) at 405 nm.

**Metastatic melanoma patients**. Lymphocytes were enriched from peripheral blood drawn from de-identified metastatic melanoma patients prior to treatment with Nivolumab or Pembrolizumab. Cells were stained immediately for flow cytometry analysis or cryopreserved for future use. CD4[+] T cells from melanoma patients were sorted or gated by CD26 expression and compared to healthy donors in parallel experiments. All patients gave written, informed consent in accordance with the Declaration of Helsinki. The Medical University of South Carolina Institutional Review Board approved this study.

**Adoptive cell transfer**. *B16F10*: B6 mice were subcutaneously injected with 4e[5] B16F10 melanoma 10 days prior to adoptive cell transfer of 5e[4] CD4[+]Vβ14[+] CD26[neg] or CD26[high] T cells. Mice received non-myeloablative 5 Gy total body irradiation 1 day pre-ACT (N = 6 mice/group, two independent experiments).

*M108*: Fig. 1: NSG mice were subcutaneously injected with 5e[6] M108 mesothelioma (50% M108 in PBS, 50% Matrigel) 40 days prior to intravenous ACT of 1e[5] redirected MesoCAR[+] human CD26[neg] or CD26[high] T cells (N = 10 mice

per group). Fig. 5: 1e[6] MesoCAR[+] CD4[+], CD26[neg], CD26[int], or CD26[high] T cells were infused into NSG mice bearing M108 tumors that were established for 60 days ($N = 7$–9 mice per group). Supplementary Fig. 6h: 5e[6] MesoCAR[+] CD4[+]CD26[high] or CD8+ T cells expanded in IL-2 were infused into M108-bearing mice and tumor regression assessed at 60 days post-ACT ($N = 6$ mice per group).

*PANC1*: NSG mice were subcutaneously injected with 5e[6] PANC1 cells (50% PANC1, 50% Matrigel) 7 days prior to intravenous ACT of 1.75e[6] CD4[+], CD26[neg], CD26[int], or CD26[high] T cells ($N = 6$–9 mice per group). All mice were measured and equally distributed among treatment groups based on tumor size. Tumors were measured bi-weekly by caliper in a blinded fashion until tumor endpoint (>400 mm[2]).

**Immunohistochemistry**. Cryosections of xenograft tumor tissues (5 μm thick) were incubated in −20 °C acetone for 10 min, rinsed and exposed to 0.3% hydrogen peroxide for 10 min before blocking and subsequent incubation with primary antibody (CD45, 2B11_PD7/26, 1:100; DAKO) overnight at 4 °C. Slides were washed and incubated with secondary antibody (Polymer-HRP) and developed with DAB substrate kit (DAKO). Slides were counterstained with hematoxylin before visualization on an Olympus BX60 microscope. IHC-stained antigen spots were counted using a computer-assisted image analyzer (Olympus Microimage Image Analysis V4.0 software for Windows). The intensities of color related to CD45 antigen spot were quantified using ImageJ and expressed as mean pixel IOD.

**Transwell migration assay**. Sorted human T cells (7.5e[4]) were re-suspended in 75 μl RPMI + 0.1% FBS and placed in the top well of a transwell plate. Chemoattractants (235 μl) were placed in the bottom well as follows: control media (RPMI + 0.1% FBS), RPMI + 10% FBS, M108 supernatant, or PANC1 supernatant. Supernatant from cancer cells was collected 15 h after plating. The ability of subsets to migrate at 37 °C for 2 h was assayed by flow cytometry.

**Western blot**. Protein was isolated and concentration quantified using a BSA Protein Assay Kit (ThermoScientific). Ten-20 μg of total protein was separated on a Mini-PROTEAN TGX, Any kDTM gel followed by transfer onto PVDF membranes (Bio-Rad). The membranes were blocked (5% non-fat dry milk in TBS + 0.5% Tween20) prior to overnight incubation at 4 °C with primary antibodies to β-catenin (BD), Bcl-2 (D17C4), Caspase-3 (8G10), or GAPDH (D16HH, Cell Signaling). Following washes, membranes were incubated for 1 h at room temperature with secondary HRP-conjugated goat antibodies to mouse or rabbit IgG (Cell Signaling). Chemiluminescence was performed using Western ECL Blotting Substrate (Bio-Rad) followed by X-ray film-based imaging. Films were scanned and quantified for integrated optical density (IOD) using ImageJ software. To remove antibodies, membranes were incubated for 15 min at room temperature in Restore Western Blot Stripping Buffer (ThermoScientific).

**Statistical analysis**. Experiments comparing two groups were analyzed using a Mann–Whitney *U* test. For multi-group comparison, a one-way analysis of variance (ANOVA) was performed with a post comparison of group using Kruskal–Wallis. Graphs utilizing error bars display the center values as the mean and error bars indicate SEM. TCRβ sequencing analysis was based on the log-linear model and the relative risks were calculated with a 95% confidence interval. For tumor curves, Mann–Whitney or Kruskal–Wallis tests were performed at the final dates where all mice from compared groups were still alive. As these experiments were exploratory, there was no estimation to base the effective sample size; therefore, we based our animal studies using traditional sample sizes ≥6.

**Data availability**. All data are available from authors upon reasonable request. OneArray data can be found at GEO accession number GSE106726.

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

## Acknowledgements

We would like to thank Hannah Knochelmann, Logan Huff and Kristina Schwartz from the Department of Microbiology & Immunology for their assistance. We also thank Adam Soloff, Zachary McPherson, Corinne Livingston and Kirsten McDaniel at the MUSC Flow Cytometry and Cell Sorting Core and Chris Fuchs at Regenerative Medicine Department Flow Core for sorting cells. We thank Kent Armeson for his guidance on statistics and acknowledge Carmine Carpenito for his assistance in comparing the M108 and PANC1 tumor models (Supplementary Fig. 6c). We thank Mark Rubinstein (PANC1), Nicholas Restifo (B16F10) and Carl June (M108, MesoCAR) for their reagents and support. Finally, we acknowledge Zihai Li, Shikhar Mehrotra, Laura Kasman, and Ramsay Camp for their constant feedback and encouragement. Finally, we acknowledge our funding sources as follows: F31 CA192787 (NCI; S.R.B.), 122709-PF-13-084-01-L1B (ACS; M.H.N.), F30 CA200272 (NCI; J.S.B.), R01 CA175061 (NCI; C.M.P.), R01 CA208514 (NCI; C.M.P.), P01 CA154778(NCI; C.M.P, M.J.Z, M.I. Nishimura) and MUSC start-up funds.

## Author contributions

S.R.B. designed and executed experiments, analyzed data, created the figures, and wrote/edited the manuscript; M.H.N., K.M., M.M.W., A.S.S. and C.C. performed experiments; M.J.Z. analyzed the data; K.S. consented and obtained blood samples from melanoma patients; C.H.J. provided reagents and intellectual feedback and C.M.P. directed the project, designed experiments, and edited the manuscript. All authors critically read and approved the manuscript.

## Additional information

**Competing interests:** S.R.B, M.H.N. and C.M.P. have a provisional patent for the use of CD26^high T cells for adoptive cell transfer therapy. The remaining authors declare no competing financial interests.

