## [Peer Review File · Nature Communications]

Reviewers' comments:

Reviewer #1 (Remarks to the Author):

1) Overall focus and significance

The study by Bailey and colleagues focusses on defining functionally distinct subsets of CD4 T cells using the CD26 marker, and comparing their potency as mediators of anti-tumor immunity. Given that the authors reference previous studies showing that CD26^{neg} T cells contain a T reg population, and that CD26^{high} cells have a Th1 or TH17 phenotype, I would concur with their suggestion that determining CD26^{int} cells are naïve is one of the main findings. However, arguably the key finding is that CD26^{high} cells are highly potent Th1/Th17-like effectors with strong potential for anti-tumour function, whose intratumoural presence correlates with better prognosis.

While there has been longstanding recognition that the strength of Th1 immunity is prognostically significant in various cancers (eg the work of Galon and colleagues in colorectal cancer), the current findings add to a number of others suggesting CD4 T cells may be very important players in anti-tumour immunity. Although some might argue that it is obvious that CD26^{neg} cells (which will be enriched for Tregs) have little/no anti-tumour capabilities, in my opinion the current study's identification of CD26^{int}/high status as a marker denoting CD4 T cell subsets that can mediate potent anti-tumour responses is one that should be of significant interest in the field, and therefore worthy of consideration for Nature Communications.

2) Is the study convincing and how could it be improved ?

The study represents a huge amount of work and overall is convincing in the conclusions it makes. The fact these conclusions are examined in both mice and human tumours and in several models, including CAR-T based cancer immunotherapy strategies, bolsters the solidity and potential significance of the findings. In addition, the robustness of the conclusions is emphasised by what appear to be appropriate statistical comparisons throughout the paper; methods appear to be adequately described.

There are however two very interesting questions that are left largely unexplored. Firstly, given the current bias of the cancer immunology field to focus on CD8 T cells as the primary anti-tumour effector cell, it would be interesting to know how the potency of the CD4 T cell subsets (eg particularly CD26^{hi} CD4 T cells) compares to that of CD8 T cell effectors. If these data are available in any of the models, they could be referred to, or if comparisons could be quickly made and included alongside that would be ideal. The text implies that in the B16F10 model, the CD26^{high} T cells are highly potent ('a mere 50,000 CD26^{high} T cells'), however a direct comparison with CD8 T cells in a relevant model would be interesting to see, and could substantially strengthen the paper.

Secondly, the study does not address the mechanism of the anti-tumour function of the CD26^{int}/high CD4⁺ T cells. Does the mechanism involve direct anti-tumour killing by CD4 T cells, or would they act indirectly, eg via promoting activation of other direct anti-tumour effectors ? If a direct mechanism is suggested, would class II MHC molecules be expressed directly on tumour cells in the models tested, and what peptide antigens would be likely targets ? Is the fact that apparently naïve (CD26^{int}) CD4 T cells exert anti-tumour capability significant in this context ? Might it imply TCR-independent effects or would these cells subsequently become primed ? It might be worth noting that previous studies have highlighted the potential of CD4 T cells to be directly cytotoxic towards cellular targets (Quezada et al, JEM, 2010; also Long-HM et al,

J.Immunol, 2011). While resolving underlying mechanisms is likely to be beyond the scope of the current manuscript, the study should at least comment more substantially on likely mechanisms in the Discussion.

3) Will the manuscript influence thinking in the field ?

The identification of a marker of anti-tumour 'effector' CD4 T cells is potentially quite significant, especially given CD26 is cell surface marker and therefore amenable to exploitation in sorting approaches for cancer immunotherapy. However if the study could also address either/both of the two questions outlined above – eg by establishing the potency of CD26high CD4 T cells relative to CD8 T cells in some of the models used (eg B16F10), or by a strengthened discussion/results about underlying mechanisms, then in my opinion it would be influential in the field.

4) Minor issues:

a) Introduction: it is unclear what the reason for focussing on CD26 was – this would be nice to introduce. In addition, this should summarise what is known about the CD26 enzymatic activity.

b) The migration assay should be explained very briefly in the main text.

c) In the first section of the Results, many of the comparisons are between CD26neg and CD26high cells, whereas it might have been nice to include CD26int cells as well.

Reviewer #2 (Remarks to the Author):

The manuscript by Bailey et al. describes how CD4+ T cells can be delineated into three distinct subsets based on their expression of the marker, CD26. The authors demonstrate clear differences at the functional level in the biology of these cells, related to their ability to elicit cytokine production, resistance to apoptosis, migration and anti-tumor activity when utilized in the adoptive transfer setting. These observations were true when applied as an adoptive cell transfer approach via TIL or via transduction with clinically-relevant chimeric antigen receptors targeting mesothelin. The manuscript and figures are very well composed. The description and characterization of individual CD26 subsets is particularly rigorous and transparent, with characterization done via multiple means including flow cytometry, gene expression and data pre/post sorting to confirm the process did not alter biomarker expression. Data is presented in a sufficient manner that would enable replication of results. Thus, the data presented, including statistical analysis are of high quality and convincing.

Overall, the data in the manuscript is clearly of high novelty, and will impact understanding of T cell intrinsic immune suppression in cancer. The conclusion that CD26 expression may simplify characterization of T helper cell subsets is supported by the data and is provocative and interesting. This has potential for a broad impact on immunotherapy, as well as T cell immunology in general.

Several minor comments below are provided that may further improve the manuscript:

1. The abstract should be modified in a manner that clarifies adoptive transfer of T cells was used in several of the experiments.

2. Introduction, 1st paragraph: It is unclear how 'discovery of cancer-specific antigens has allowed for the advancement of immune checkpoint blockades'. This statement may certainly apply to advancing adoptive cell therapy, but not necessarily immune checkpoint blockade. This should be

corrected, or clarified supported by specific evidence before making these claims.

3. Introduction, 2nd paragraph: specify "immune" checkpoint modulators, rather than "checkpoint modulators" to not confuse with modulation of cell cycle checkpoints.

4. Results, Did the authors apply a different order of gating to cells to determine what percentage of CD4+CD25+FoxP3+ T regs also express CD26 at various levels? This may be of interest and could be mentioned in the results.

5. Although the manuscript does an outstanding job at exploring the expression of CD26 on T cells in the setting of adoptive transfer, it would be informative to know whether or not the authors have looked at the distribution of CD26 expression on tumors that have not undergone any therapy. Are they predominantly CD26 low? It is recommended that an experiment be done whereby tumors are grown, TILs extracted and phenotyped to detect whether the different CD26 expressing CD4+ T cell subsets are present in tumors. This could extend significance of the data beyond the adoptive cell therapy setting.

6. Discussion: Can the authors comment on whether they think the expression of CD26 on cells is fluid/dynamic event, or a terminal event in a CD4+ T cell? For example, it may be worth noting any exogenous factors or stimuli that are reported in the literature to upregulate CD26 expression on a cell. These type of data may be relevant for future studies in understanding the regulation of this marker and its significance in tumor-immune interactions.

7. Results: Spelling: change "phenomena" to "phenomenon" in the "CD26 high cells are multi-functional and enzymatically active" paragraph.

8. Results: In title of paragraph specify peripheral blood was examined by changing title to read as follows: "CD26 negative, intermediate and high T cells exist in peripheral blood of cancer patients".

9. Results: The data looking at adoptive transfer of T cells in pancreatic tumors shows a clear inhibition of tumor growth rate. However, this reviewer takes issue with referring to this as 'regression'. It is over-stated to say that CD26^{int} and CD26^{high} T cells "regressed" pancreatic tumors. These conclusions should be adjusted throughout the manuscript to more accurately reflect the data.

10. Discussion/Results: the authors convincingly suggest that CD26 expression can be used to simplify phenotyping of T cells. This is very attractive and innovative conceptually that is certainly supported by data. They further state that "several markers such as CD45RA, CCR7, CD62L and CD95" can be consolidated. This may be true, and they present very supportive data for CD45RA, CCR7 and CD62L in the manuscript. However, data on CD95 is lacking in the manuscript. Do they also have data on CD95 to support this claim? If so, it should be shown at least as a supplemental figure.

11. Discussion: The sentence in the 2nd paragraph is incomplete and should be revised: "... infection post-serial transfer (REF41), would be interesting..." I believe it should read "it would be interesting."

12. Methods: under microarray correct spelling of "Qiagen".

13. Methods: please confirm whether or not metastatic melanoma patients were receiving any therapy (particularly immunotherapy) at the time of blood draw. This could influence interpretation of results if receiving therapy.

14. Supplementary Figure 1. The title does not reflect the human data contained in the latter part

of the figure. Would recommend either changing title or splitting into 2 separate figures. Also the abbreviation "NT" presumably 'not treated' is not spelled out in the legend. Finally, clarify in the legend that in (F) these are CD4+ T cells.

15. Supplementary Figure 7. It may improve data presentation to adjust the y-axis in Figure S7B to a maximum value of 10 or 20 to better visualize significant differences in the data. As it stands it looks crunched.

Dear Reviewers,

We appreciate the time and effort put forth to review our submitted manuscript entitled “**Human CD26^{high} T cells elicit tumor immunity against multiple malignancies via enhanced migration and persistence**” (NCOMMS-17-05607). Please find our point-by-point responses (**bold text**) to your requested revisions/comments (plain text) below. Note that we have included the two reviewers comments verbatim. All changes in the manuscript have also been highlighted **in yellow** for easier viewing. Your feedback has considerably strengthened this manuscript and we are grateful for your suggestions.

We hope that these revisions meet your expectations and appreciate the chance to resubmit our work for publication in *Nature Communications*.

Best regards,

Chrystal Paulos, PhD

Stefanie Bailey

In response to Reviewer #1:

First, we'd like to thank you for your interest in our work and the belief that these findings are “potentially quite significant” and “should be of significant interest in the field”. We appreciate your insight and have chosen to respond to your queries as follows:

1. “Firstly, given the current bias of the cancer immunology field to focus on CD8 T cells as the primary anti-tumour effector cell, it would be interesting to know how the potency of the CD4 T cell subsets (eg particularly CD26^{hi} CD4 T cells) compares to that of CD8 T cell effectors. If these data are available in any of the models, they could be referred to, or if comparisons could be quickly made and included alongside that would be ideal. The text implies that in the B16F10 model, the CD26^{high} T cells are highly potent “a mere 50,000 CD26^{high} T cells???”), however a direct comparison with CD8 T cells in a relevant model would be interesting to see, and could substantially strengthen the paper.”

This is a great question and we chose to address this request in two parts. First, we assayed human MesoCAR⁺ CD4⁺CD26^{high} and CD8⁺ T cells *in vitro* using a cytotoxic assay, which permit us to determine the ability of these subsets to kill cancer. Second, we infused these human, tumor-specific subsets into mesothelioma-bearing mice to assess their antitumor capacity *in vivo*. Interestingly, we found that while both subsets had similar cytotoxicity *in vitro*, CD4⁺CD26^{high} T cells were significantly better at regressing tumors *in vivo* than

their CD8⁺ counterparts, thus underscoring that these “helper” cells have a potent cytotoxic potential. Our resulting data can be found in Supplementary Fig. 6G-I.

2. “Secondly, the study does not address the mechanism of the anti-tumour function of the CD26^{int}/high CD4⁺ T cells. Does the mechanism involve direct anti-tumour killing by CD4 T cells, or would they act indirectly, eg via promoting activation of other direct anti-tumour effectors? If a direct mechanism is suggested, would class II MHC molecules be expressed directly on tumour cells in the models tested, and what peptide antigens would be likely targets? Is the fact that apparently naive (CD26^{int}) CD4 T cells exert anti-tumour capability significant in this context? Might it imply TCR-independent effects or would these cells subsequently become primed? It might be worth noting that previous studies have highlighted the potential of CD4 T cells to be directly cytotoxic towards cellular targets (Quezada et al, JEM, 2010; also Long-HM et al, J.Immunol, 2011). While resolving underlying mechanisms is likely to be beyond the scope of the current manuscript, the study should at least comment more substantially on likely mechanisms in the Discussion.”

We agree with the reviewer that while it would be interesting to discern the mechanism of action of these subsets, it does lie outside of the scope of this paper. However, we have made two additions to address your question. First, we’ve clarified that since our NSG mouse models are only receiving CD4⁺ T cells and have no inherent immune system, our findings suggest that the human CD4⁺ subsets are directly clearing tumor themselves (pg. 14). Furthermore, the use of the CAR model removes any question on what MHC II molecules and tumor antigens are important for this finding as this synthetic antigen receptor, in contrast to traditional TCRs, recognizes full mesothelin on M108 tumors. Second, we’ve included an *in vitro* cytotoxic assay (Supplementary Fig. 6G, mentioned above) that shows that CD4⁺CD26^{high} T cells are capable of killing cancer cells *in vitro* 1) by themselves and 2) comparably, if not better, than CD8⁺ T cells. Collectively, we believe that this data shows that human CD4⁺CD26^{high} T cells can directly eradicate tumors. However, in future studies, we will seek to address how distinct functional properties of CD26^{high} T cells regulate immune responses to cancer to shed light on the exact killing mechanism of this lymphocyte population.

3. “Introduction: It is unclear what the reason for focusing on CD26 was, this would be nice to introduce. In addition, this should summarize what is known about the CD26 enzymatic activity.”

We agree and apologize for being vague. A paragraph on why CD26 was chosen to investigate has now been included in the introduction on pages 4-5. Briefly, we sought to study this important molecule due to the previously published correlation between CD26 expression and IL-17 production. Given that Th17 cells have been shown to have stem cell-like qualities, we sought to determine if CD26 expression would correlate with a long-lived cell to enhance ACT

therapy. Although some of the original introduction was removed to make room for this statement, there was still limited space to summarize the current literature on enzymatic activity. This information was briefly touched on, but is more defined in both the results and discussion sections of the paper.

4. "The migration assay should be explained very briefly in the main text."

A brief assay description has been added to the main text on page 14.

5. "In the first section of the Results, many of the comparisons are between CD26neg and CD26high cells, whereas it might have been nice to include CD26int as well."

While this is a logical point and we do have the data for CD26^{int} T cells to complete this section, we do not wish to include this data in the figure. Due to the naïve properties of CD26^{int} T cells, we believe that the way we have presented our findings of how unique the CD26^{high} T cells are due to their ability to control tumor despite their differentiated phenotype is strengthened by introducing these cells later in the document. We think that comparing T cells with different levels of CD26 expression upfront disrupts the flow of the paper and distracts from the overall findings.

In response to Reviewer #2:

We greatly appreciate your feedback and are glad that you agree with us that this work is "of high novelty" and "has potential for a broad impact on immunotherapy". We are grateful for your comments and have chosen to respond as follows:

1. "The abstract should be modified in a manner that clarifies adoptive transfer of T cells was used in several of the experiments."

We agree and ACT is now stated in the abstract. Thank you!

2. "Introduction, 1st paragraph: It is unclear how "discovery of cancer-specific antigens has allowed for the advancement of immune checkpoint blockades". This statement may certainly apply to advancing adoptive cell therapy, but not necessarily immune checkpoint blockade. This should be corrected, or clarified supported by specific evidence before making these claims.

Good point. This statement on page 4, paragraph 1 has been modified to increase clarity.

3. "Introduction, 2nd paragraph: specify "immune checkpoint modulators" rather than "checkpoint modulators" to not confuse with modulation of cell cycle checkpoints."

We agree. The wording of this sentence has been changed as requested.

4. "Results, Did the authors apply a different order of gating to cells to determine what percentage of CD4+CD25+FoxP3+ T regs also express CD26 at various levels? This may be of interest and could be mentioned in the results"

Good question. This data has been added as Supplementary Fig. 1F and discussed briefly in the results section on page 6. We found that natural Tregs (FoxP3⁺Helios⁺) were mainly devoid of CD26 expression (~88%), confirming that CD26 is lacking on regulatory T cells.

5. “Although the manuscript does an outstanding job at exploring the expression of CD26 on T cells in the setting of adoptive transfer, it would be informative to know whether or not the authors have looked at the distribution of CD26 expression on tumors that have not undergone any therapy. Are they predominantly CD26 low? It is recommended that an experiment be done whereby tumors are grown, TILs extracted and phenotyped to detect whether the different CD26 expressing CD4⁺ T cell subsets are present in tumors. This could extend significance of the data beyond the adoptive cell therapy setting.”

This was an interesting question and has greatly improved the quality of the paper. To answer this question, we performed an experiment in mice bearing B16F10 melanoma tumors that received no therapeutic intervention. This new data is now displayed in Supplemental Fig. 6F and mentioned in the results section on page 13. While the question simply asks if the CD26 subsets are present on endogenous TIL, we found that these cells are not only present in tumor, but they also correlate with tumor size similar to that of mice receiving ACT therapy. Specifically, we found that CD26 expression on endogenous TIL was reduced in mice with larger tumors. We found this very exciting/interesting because it suggests a correlation between CD26 expression on CD4⁺ T cells and their subsequent ability to clear tumor.

6. “Discussion: Can the authors comment on whether they think the expression of CD26 on cells is fluid/dynamic event, or a terminal event in a CD4⁺ T cell? For example, it may be worth noting any exogenous factors or stimuli that are reported in the literature to upregulate CD26 expression on a cell. These type of data may be relevant for future studies in understanding the regulation of this marker and its significance in tumor-immune interactions.”

Due to limited space, we were unable to address this key comment to the depth that we would have liked. However, we did include a sentence in the discussion on page 18 regarding the up-regulation of CD26 following activation. We seek to address this question in more depth in future manuscripts as we agree with you that the regulation of CD26 on tumor-specific T cells might be very informative in understanding, and perhaps improving, cancer immunotherapy.

7. “Results: Spelling: change ‘phenomena’ to ‘phenomenon’ in the CD26high cells are multifunctional and enzymatically active paragraph.”

We apologize. This spelling error on page 10 has now been corrected.

8. "Results: In title of paragraph specify peripheral blood was examined by changing title to read as follows: "CD26 negative, intermediate and high T cells exist in peripheral blood of cancer patients"?"

This title on page 10 has been revised as requested.

9. "Results: The data looking at adoptive transfer of T cells in pancreatic tumors shows a clear inhibition of tumor growth rate. However, this reviewer takes issue with referring to this as "regression". It is over-stated to say that CD26int and CD26high T cells "regressed" pancreatic tumors. These conclusions should be adjusted throughout the manuscript to more accurately reflect the data."

We agree. All statements regarding pancreatic tumor results throughout the manuscript have been revised to clarify that tumors were slowed/inhibited growth rate instead of regressed.

10. "Discussion/Results: the authors convincingly suggest that CD26 expression can be used to simplify phenotyping of T cells. This is very attractive and innovative conceptually that is certainly supported by data. They further state that "several markers such as CD45RA, CCR7, CD62L and CD95" can be consolidated. This may be true, and they present very supportive data for CD45RA, CCR7 and CD62L in the manuscript. However, data on CD95 is lacking in the manuscript. Do they also have data on CD95 to support this claim? If so, it should be shown at least as a supplemental figure."

CD95 data has been added as requested. Please see Supplementary Fig. 2C, which shows that CD95 is highly expressed on CD26^{high} T cells.

11. "Discussion: The sentence in the 2nd paragraph is incomplete and should be revised: ...infection post-serial transfer (REF41), would be interesting. I believe it should read 'it would be interesting'."

After further observation of this statement, we do not agree that it should be changed from its current state. We believe that "Studies similar to that by Graef et. al, ..., would be interesting to investigate..." is the correct wording.

12. "Methods: Under microarray correct spelling of Qiagen."

Done. Nice catch ;)

13. "Methods: Please confirm whether or not metastatic melanoma patients were receiving any therapy at the time of blood draw."

We apologize for the confusion. Patient samples were collected prior to the start of checkpoint inhibitor therapy. This information has been added to the Methods section on page 23.

14. "Supplementary Figure 1. The title does not reflect the human data contained in the latter part of the figure. Would recommend either changing title or splitting into two separate figures."

We agree. This title has been revised so that it does not denote simply human or mouse data and now includes both the activation and tumor regression aspects of the figure.

15. "Supplementary Figure 1. ...the abbreviation NT is not spelled out in the legend."

Thanks. NT (No Treatment; untreated) has now been defined in the legend.

16. "Supplementary Figure 1. Clarify in the legend that in (F) these are CD4⁺ T cells."

Clarification on the use of CD4⁺ T cells has been added to the legend.

17. "Supplementary Figure 7. It may improve data presentation to adjust the y-axis in Figure S7B to a maximum value of 10 or 20 to better visualize significant differences in the data. As it stands it looks crunched."

We agree and the figure axis has been augmented as requested.

REVIEWERS' COMMENTS:

Reviewer #1 (Remarks to the Author):

I am broadly happy with the response to my queries, although I would like the authors to speculate on the mechanism by which non-modified CD26+ CD4 T cells (eg as components of endogenous TIL populations) might mediate anti-tumour effects (eg cytokine production (IFN-g, IL-17??, or direct cytotoxicity, importance of class II MHC etc). I note that transcriptional studies of human colorectal cancer have highlighted that human CD4 and class II MHC expression inside the tumour correlates more closely with the strength of intratumoural immunity than CD8 (Lal et al, Oncoimmunology, 2015), and the mechanism by which they may act is an important consideration, particularly in light of the findings that adoptive transfer of single class II MHC-restricted clones can apparently mediate potent and clinically significant anti-tumour immunity (Tran et al, Science, 2014).

An immunogenomic stratification of colorectal cancer: Implications for development of targeted immunotherapy.

Lal N, Beggs AD, Willcox BE, Middleton GW.

Oncoimmunology. 2015 Apr 2;4(3):e976052. eCollection 2015 Mar.

Science. 2014 May 9;344(6184):641-5. doi: 10.1126/science.1251102.

Cancer immunotherapy based on mutation-specific CD4+ T cells in a patient with epithelial cancer.

Tran E1, Turcotte S, Gros A, Robbins PF, Lu YC, Dudley ME, Wunderlich JR, Somerville RP, Hogan K, Hinrichs CS, Parkhurst MR, Yang JC, Rosenberg SA.

Reviewer #2 (Remarks to the Author):

All comments have been addressed from my prior review. The authors actually went above and beyond to address the comments from each of the two reviews with new experimentation, greater clarity of presentation and a thoughtful, well-organized point-by-point response that allowed for efficient review of the work.

Only one minor spelling error should be corrected during the editorial process:
line 574 should read "Pembrolizumab".

Dear Reviewers,

We once again appreciate the time and effort put forth to review our resubmitted manuscript entitled “**Human CD26^{high} T cells elicit tumor immunity against multiple malignancies via enhanced migration and persistence**” (NCOMMS-17-05607). Please find our point-by-point responses (**bold text**) to your requested revisions/comments (plain text) below. Note that we have included the two reviewers comments verbatim. All changes in the manuscript can be identified by the ‘tracked changes’ for easier viewing. Your feedback has once again improved this manuscript and we are grateful for your suggestions.

We hope that these revisions meet your expectations and appreciate the chance to resubmit our work for publication in *Nature Communications*.

Best regards,

Chrystal Paulos, PhD

Stefanie Bailey

In response to Reviewer #1:

1. “I am broadly happy with the response to my queries, although I would like the authors to speculate on the mechanism by which non-modified CD26+ CD4 T cells (eg as components of endogenous TIL populations) might mediate anti-tumour effects (eg cytokine production (IFN-g, IL-17??, or direct cytotoxicity, importance of class II MHC etc). I note that transcriptional studies of human colorectal cancer have highlighted that human CD4 and class II MHC expression inside the tumour correlates more closely with the strength of intratumoural immunity than CD8 (Lal et al, Oncoimmunology, 2015), and the mechanism by which they may act is an important consideration, particularly in light of the findings that adoptive transfer of single class II MHC-restricted clones can apparently mediate potent and clinically significant anti-tumour immunity (Tran et al, Science, 2014).”

This is an excellent point and although we had a brief speculation in the original discussion, we have significantly built on this section to better address this request. As you’ll see in the discussion section, we now speculate on how CD4⁺CD26⁺ T cells can be altering antitumor immunity in both MHC-independent and dependent ways. First, we believe that our CAR tumor models show that these cells have an MHC-independent capacity to elicit immunity that could be due to the following: 1) Enhanced multifunctionality (i.e. IFN γ , Granzyme B, etc.), 2) Greater Teff to Treg ratio and 3) Heightened

activation of TIL. Given our similar findings in TCR-specific and endogenous TIL, we speculate that these cells could also have an MHC-dependent role in tumor control. Due to the known expression of CD26 on memory T cells, as well as the similar phenotype of our CD26^{high} T cells to those presented by Tran et. al in metastatic cholangiocarcinoma, it's highly possible that CD26 expression correlates with antigen-specific T cells that could be utilized in future TCR therapies. I thoroughly enjoyed researching this topic to generate the revisions and I genuinely believe that it strengthened the paper. Thank you for advising us to further expand on this concept.

2. "All comments have been addressed from my prior review. The authors actually went above and beyond to address the comments from each of the two reviews with new experimentation, greater clarity of presentation and a thoughtful, well-organized point-by-point response that allowed for efficient review of the work.

Only one minor spelling error should be corrected during the editorial process:
line 574 should read "Pembrolizumab"."

Thank you for bringing this to our attention. The correction has been made as requested. Your attention to spelling and detail has without a doubt improved the quality of this manuscript.